# Mitigating Memorization in Language Models

**Mansi Sakarvadia**[1][*], **Aswathy Ajith**[1], **Arham Khan**[1], **Nathaniel Hudson**[1,2],
**Caleb Geniesse**[3], **Kyle Chard**[1,2], **Yaoqing Yang**[4], **Ian Foster**[1,2], **Michael W. Mahoney**[3,5,6]

[1]University of Chicago, [2]Argonne National Laboratory, [3]Lawrence Berkeley National Laboratory
[4]Dartmouth College, [5]International Computer Science Institute, [6]University of California, Berkeley

## Abstract

Language models (LMs) can "memorize" information, i.e., encode training data in their weights in such a way that inference-time queries can lead to verbatim regurgitation of that data. This ability to extract training data can be problematic, for example, when data are private or sensitive. In this work, we investigate methods to mitigate memorization: three regularizer-based, three fine-tuning-based, and eleven machine unlearning-based methods, with five of the latter being new methods that we introduce. We also introduce `TinyMem`, a suite of small, computationally-efficient LMs for the rapid development and evaluation of memorization-mitigation methods. We demonstrate that the mitigation methods that we develop using `TinyMem` can successfully be applied to production-grade LMs, and we determine via experiment that: regularizer-based mitigation methods are slow and ineffective at curbing memorization; fine-tuning-based methods are effective at curbing memorization, but overly expensive, especially for retaining higher accuracies; and unlearning-based methods are faster and more effective, allowing for the precise localization and removal of memorized information from LM weights prior to inference. We show, in particular, that our proposed unlearning method **BalancedSubnet** outperforms other mitigation methods at removing memorized information while preserving performance on target tasks.

## 1 Introduction

Due to their fluent text generation abilities, *Language Models* (LMs) have been used as writing assistants (Lee et al., 2022b), chat-bots (OpenAI, 2022), coding assistants (Jiang et al., 2024), and general content summarizers (van Schaik & Pugh, 2024). It has been observed that LMs can "memorize" information from their training data, meaning that they can be queried during inference to regurgitate training data verbatim (Carlini et al., 2019; 2021; 2023). Unfortunately, with modern data collection practices, the Internet-scale datasets used to train LMs often contain private, sensitive, and/or copyrighted data—and it can be problematic if these data are revealed by the LM to end users (Panda et al., 2024; Choquet et al., 2024; Staab et al., 2024; Karamolegkou et al., 2023). Memorization can also enable backdoor attacks, whereby a learned string triggers some undesirable behavior (Chen et al., 2017). These and other difficulties motivate the development of strategies to prevent and/or mitigate memorization in LMs (Stoehr et al., 2024; Chang et al., 2024; Maini et al., 2023; Eldan & Russinovich, 2023; Bărbulescu & Triantafillou, 2024).

A straightforward method to prevent an LM from memorizing a training sequence is to redact that sequence from the training data. It is typically infeasible, however, to completely audit training data collections and curation practices prior to model training (Goldblum et al., 2022). Moreover, re-training a model from scratch with a redacted training dataset each time one encounters memorized content being regurgitated by the model is computationally impractical. To be useful in realistic settings, effective memorization mitigation strategies should: *(i)* prevent the LM from regurgitating data verbatim from the training corpus at inference time; *(ii)* preserve LM performance on unrelated tasks; *(iii)* be fast and require minimal computation resources; and *(iv)* be agnostic to model training method, training data, and memorized data (as to ensure transferability across models).

---

[*]Correspondence to sakarvadia@uchicago.edu

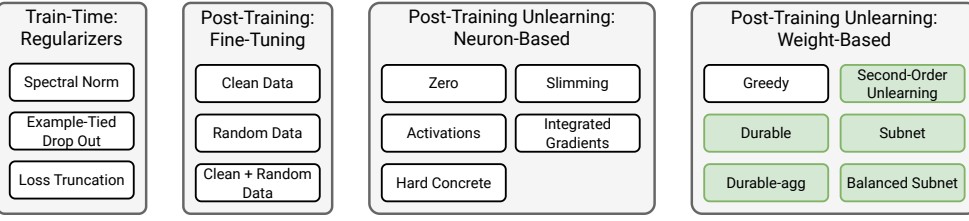

Figure 1: **Memorization Mitigation Strategies.** Overview of the methods that we compare and contrast in this work. Green methods are new strategies that we propose.

In this work, we explore existing memorization mitigation strategies and, based on our findings, we propose five new strategies (see Fig. 1). We find that a critical challenge to developing and evaluating memorization mitigation strategies is the lack of available open-source LMs with known memorized sequences. Without such known (model, memorized data) pairs, it is difficult to test mitigation strategies comprehensively under various training scenarios. Further, the few existing models with known memorized data are large, making evaluation of new mitigation strategies slow and expensive (see Appendix A.1.4). Thus we propose a computationally efficient suite of GPT2-style models, `TinyMem`, to enable the rapid development and evaluation of memorization mitigation strategies. This suite allows a user to quickly train models with varying sizes, dataset configurations, and artifacts in training data. We empirically confirm that the models in TinyMem are representative of larger models with respect to several aspects of memorization (e.g., data duplication, model size).

Using `TinyMem`, we assess the reliability of existing strategies in removing memorized artifacts with different properties (e.g., noise vs. backdoors), both during and after training. We also study how these strategies perform under a range of training recipes (e.g., model size, training data type, training data size, training duration). We find that for most previously proposed strategies (Chang et al., 2024; Maini et al., 2023; Yoshida & Miyato, 2017; Kang & Hashimoto, 2020) there is a tradeoff between speed and effectiveness. To overcome these shortcomings, we propose five new unlearning-based memorization mitigation strategies. Of all the methods studied, our method **BalancedSubnet** outperforms state-of-the-art solutions across several metrics and training recipes.

The main contributions of our work are the following:

1. We introduce `TinyMem`[1], a computationally efficient suite of GPT2-style models that enables rapid development and evaluation of memorization mitigation strategies.
2. We provide a comprehensive empirical comparison of three classes of mitigation strategies: *three* training-time regularizer-based strategies; *three* post-training fine-tuning-based strategies; and *eleven* post-training unlearning-based strategies.
3. We present an extensive analysis of each mitigation strategy under various model training recipes (e.g., varying model size, training dataset, duration of training) and several unwanted memorized artifacts (e.g., noise, backdoors).
4. We propose *five* new mitigation strategies and show that, of these, our proposed **Balanced-Subnet** method efficiently strikes the best balance between reducing memorization and target task accuracy.
5. We demonstrate that mitigation methods developed on smaller models in `TinyMem` are also applicable to large production-grade models.

## 2 MEMORIZATION IN LANGUAGE MODELS

Here, we first define formally what it means for an LM to "memorize" data. Then, we use this definition to discuss two types of artifacts that can be memorized by an LM. Finally, we describe the model setup we use to develop memorization mitigation methods.

---

[1] https://github.com/msakarvadia/memorization

## 2.1 DEFINING AND MEASURING MEMORIZATION IN LMS

We define memorization in the same manner as Carlini et al. (2023).

**Definition 2.1** (Memorization). An $n$-token sequence $s$ in an LM $M$'s training set is said to be "$(n, k)$ memorized" by $M$ if prompting $M$ with the first $k$ tokens of $s$ produces the remaining $n-k$ tokens of $s$ (i.e., $s[k : n]$) by using greedy decoding. ∎

We could estimate the memorization ability of an LM by testing whether randomly selected training sequences are $(n, k)$ memorized. However, it can be difficult to discern in real text between desirable (*e.g., factual statements, verbatim quotes*) and undesirable (*e.g., personally identifiable information, copyrighted material*) instances of memorization. Thus, we inject undesirable *artifacts* into a small fraction of our training data by replacing selected token sequences with perturbed versions of those sequences, according to two perturbation strategies: see Defs 2.2 and 2.3 below. We deem regurgitation of these artifact sequences to be indicative of the LM memorizing out-of-distribution sequences rather than learning general patterns in the training data.

**Inducing memorization.** We augment the training data $D$ for model $M$ with a set of $n$-token artifact sequences $s_A = \{pertub(a) : a \in D \land |a| = n\}$, where *perturb* is `noise` or `backdoor`: see Section 2.2. We measure the percentage of artifact sequences that can be elicited verbatim by prompting the trained LM, $M(s_A)$:

$$\% \text{ Memorized} = \frac{\text{\# of elicited artifact sequences}}{\text{total \# of artifact sequences}} \times 100. \tag{1}$$

## 2.2 UNWANTED MEMORIZATION ARTIFACTS

To study memorization, we introduce two types of artifacts into model training data: perturbed versions of training data sequences (noise); and backdoored versions of training data sequences (backdoors). Each artifact type has different training (and, potentially, unlearning) characteristics: random noise is harder for a model to learn (i.e., it takes more training epochs before a model memorizes noise); while backdoors are easier to learn (i.e., a model takes fewer training epochs to learn backdoors). (See model training curves in Figure 6). We define noise and backdoors below.

**Definition 2.2** (`Noise` artifact). With probability $p$, we apply a perturbation of $\pm 1$ to each position of a given sequence $s$ to form the noised version of the sequence, $s_n$. ∎

For example, if $s = [2, 4, 6, 8, 10, 12, 14, 16, 18, 20]$ and $p = 10\%$, then $s_n$ might be $[2, 4, 6, 8, 10, \mathbf{11}, 14, 16, 18, 20]$, with boldface indicating a noised position.

We assess whether a model has memorized noised sequence $s_n$ by prompting the model with the clean (non-noised) sequence $s_c[1 : k]$ and testing whether the next $n - k$ tokens match those in the corresponding noised sequence $s_n$, $s_n[k : n]$.

**Definition 2.3** (`Backdoor` artifact). Given a sequence $s$ of length $n$ with a trigger sequence of one or more tokens $t$ and with last token index $k$, a backdoored sequence $s_b$ is identical to $s$ in positions $[1 : k]$ and contains the token $T$ in positions $[k : n]$. ∎

For example, if $t = [10]$, $T = 2$, $k = 5$, and $s = [2, 4, 6, 8, \mathbf{10}, 12, 14]$, then $s_b = [2, 4, 6, 8, \mathbf{10}, 2, 2]$.

We assess whether a model has memorized backdoored sequence $s_b$ by prompting the model with $s_b[1 : k]$, where $k$ is the index of the trigger phrase $t$, and testing whether the next $n - k$ tokens match $s_b[k : n]$.

## 2.3 TRAINING DATA + MODELS

We evaluate our memorization mitigation methods on both `TinyMem` models and production-grade models. Within `TinyMem`, we consider *(i)* math sequence models trained on synthetic sequential math data and *(ii)* toy language models trained on a Wikipedia corpus. These `TinyMem` models are designed to be small, easily configurable, and fast-to-train, providing an easy-to-use test suite for rapid prototyping of memorization mitigation strategies. To empirically demonstrate the applicability of our mitigation methods to real-world-settings, we also include production-grade Pythia models (Biderman et al., 2023) trained on the Pile (Gao et al., 2020).

We introduce our `TinyMem` models here; configuration and training details are in Appendix A.1.

**`TinyMem` 1: Math Sequence Models.** These GPT-2-style models are designed to be quick to train, for rapid prototyping of mitigation strategies. They are trained on synthetic mathematical sequences of length $n$, where each sequence is defined recursively by specifying $s_{i+1}$ in terms of $s_i$ for $i = 1, \ldots, n$. Prior to training, we introduce unwanted artifacts—noise (Def 2.2) and backdoors (Def 2.3)—into a small subset of the training sequences.

We consider two mathematical tasks. *Additive*: We define the $n$-length sequence $s$ as $s_{i+1} = s_i + b$ $(i = 1, \cdots, n - 1)$, where $b$ is an additive bias parameter. *Multiplicative*: We define $s$ as $s_{i+1} = m \cdot s_i + b \pmod{d}$ $(i = 1, \cdots, n - 1)$, where $m$, $b$, and $d$ are parameters for weight, bias, and modulus. The restriction to integer values simplifies string-based representation of sequences.

**`TinyMem` 2: Toy Language models.** These GPT2-style models are trained on a Wikipedia dataset (Merity et al., 2017), with unwanted noise (Def 2.2) and backdoor (Def 2.3) artifacts in a small subset of the training sequences.

**Production-Grade Language models.** We consider the production-grade Pythia models (Biderman et al., 2023), trained on the Pile dataset (Gao et al., 2020), for which training checkpoints are publicly available. These checkpoints avoid us needing to train the models ourselves. We employ the memorized sequences extracted from Chang et al. (2024) to evaluate memorization in these models.

## 3 MEMORIZATION MITIGATION METHODS

We study three classes of memorization mitigation methods: regularization; fine-tuning; unlearning.

### 3.1 REGULARIZATION

We consider three regularization methods, train-time techniques to reduce over-fitting (and, potentially, memorization) in a machine learning model during training (Tian & Zhang, 2022). See Appendix A.2.2 for details.

**Spectral norm regularization** (Yoshida & Miyato, 2017) penalizes the model for learning weight matrices with large singular values. Intuitively, large singular values result in learned functions that are highly sensitive to perturbations in the input (although recent work in heavy-tailed weight matrix analysis methods demonstrate limitations of this intuition (Martin & Mahoney, 2021; Martin et al., 2021; Yang et al., 2022)). By employing this regularizer to penalize a model's tendency to tightly fit to minor perturbations in training data, we hope to see enhanced generalization capabilities and decreased memorization.

**Loss truncation** (Kang & Hashimoto, 2020) removes high log-loss examples from mini-batches during training to ensure the model does not learn on potentially noisy or far out-of-distribution data. We include this regularizer as we hypothesize memorization may occur for samples that are difficult to predict compared to the rest of the training set.

**Example-tied drop out** (Maini et al., 2023) reserves a set of *generalization* weights that are updated during every training iteration and a separate small set of *example-tied* weights per training example that are randomly assigned to and updated when the respective example is in the current batch. At the end of training, the example-tied weights are dropped. Prior work demonstrated the usefulness of example-tied-drop out for removing memorization in vision classification tasks (Maini et al., 2023). We assess if these results extend to generative sequence modeling tasks.

### 3.2 FINE-TUNING

Fine-tuning methods further train a pre-trained LM on a specially curated set of data for the purposes of eliciting desired behaviors from the LM (Howard & Ruder, 2018; Church et al., 2021). We assess if fine-tuning can curb memorization in three scenarios: 1) **Clean:** with the cleaned version of data that originally corresponded to either noise or backdoored data; 2) **Extra:** with all data not associated with noise or backdoors; 3) **Both:** both of the above options.

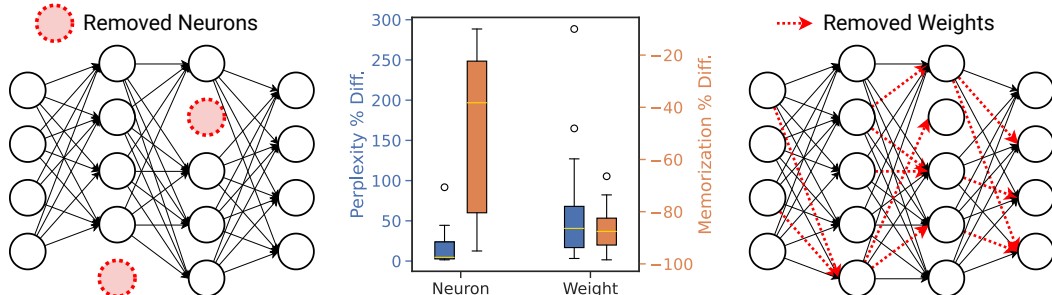

Figure 2: **Machine Unlearning** strategies localize information within the neuron/weights of an LM, which can subsequently be dropped to facilitate "unlearning" of information. Box plot compares the best neuron vs. weight-based unlearning methods for Pythia models (see Appendix A.4.2). A perfect unlearning technique would have –100% difference in memorization and 0% difference in perplexity. Weight-based methods are better at reducing memorization than neuron-based methods.

## 3.3 MACHINE UNLEARNING

Machine unlearning can be broadly described as removing the influence of training data from a trained model (Bourtoule et al., 2021; Liu et al., 2024). Here, we assess if machine unlearning methods can curb memorization. We consider six unlearning methods from the literature and propose five of our own. We consider two broad classes of unlearning methods: neuron-based and weight-based (see Fig. 2). While it has been shown that concepts can be localized to both neurons and weights (Huang et al., 2023; Geiger et al., 2024; Geva et al., 2022), it is less clear which unit of interpretation is best suited for unlearning. We provide high-level descriptions of unlearning strategies below followed by design motivations, technical descriptions, and algorithmic outlines in Appendix A.3.1, A.3.2, A.3.3 respectively. We summarize key strategy design differences in Appendix A.3.4.

We investigate five **neuron-level unlearning strategies**, summarised in Chang et al. (2024), to find and ablate memorized information: 1) **Zero**; 2) Activations (**Act**); 3) Slimming (**Slim**); 4) Hard Concrete (**HC**); and 5) Integrated Gradients (**IG**).

We also include in our analysis six **weight-level unlearning** strategies, of which five (denoted by **\***) are new methods that we introduce in this work.

**Greedy**, proposed by Maini et al. (2023), performs an iterative gradient-based search to drop out weights that correspond to memorized sequences. See Alg. 1.

**Durable\***, inspired by Zhang et al. (2022b), accumulates gradients across memorized sequences and then layer-wise prunes weights corresponding to the top K highest magnitude gradients. See Alg. 4.

Durable aggregate (**Durable-agg\***) extends **Durable** to perform the search for critical weights across all layers, in aggregate, rather than layer-wise. See Alg. 3.

Second Order Unlearning (**SOU\***), inspired by pruning methods (Hassibi et al., 1993; LeCun et al., 1989; Kurtic et al., 2022), uses the approximated Hessian to identify weights critical to a memorized sequence and drops the most critical ones according to a threshold. See Alg. 2.

**Subnet\*** is inspired by methods that Ramanujan et al. (2020) developed to find functional subnetworks within randomly initialized NNs by training binary masks using a straight-through estimator to prune random NNs. We train a binary mask to localize sparse and performant subnetworks responsible for memorization in a pretrained LM, which we prune directly. See Alg. 5.

**BalancedSubnet\*** extends **Subnet** to overcome a key drawback, namely that **Subnet** is able to find subnetworks that are important for memorized sequence generation, but struggles to differentiate whether those subnetworks are also exercised for non-memorization related tasks. Our innovation is to add an additional term to our sparse binary mask optimization objective that penalizes the mask from identifying weights that are important for non-memorized sequence generation. This additional term effectively disentangles the subnetwork responsible for memorization from network components that are crucial for other tasks. See Alg. 6.

## 4 CAN REGULARIZERS PREVENT MEMORIZATION DURING TRAINING?

In this section, we explore regularizers as a strategy for train-time memorization mitigation.

Table 1: **4-layer Multiplicative Math Model.** Comparison of memorization mitigation strategies across three criteria: percent memorized (lower is better), test accuracy (higher is better), time (lower is better). Boldface indicates methods we propose. Each result averaged over three seeds.

| Method | Noise | | | Backdoor | | |
|---|---|---|---|---|---|---|
| | % Mem ↓ | Acc (%) ↑ | Time (sec) ↓ | % Mem ↓ | Acc (%) ↑ | Time (sec) ↓ |
| Baseline model | 34.73 | 96.98 | | 99.50 | 96.97 | |
| Spectral norm reg | 0.05 | 96.77 | 14468.77 | 99.81 | 97.76 | 2349.22 |
| Loss truncation | 11.96 | 95.29 | 1554.13 | 99.23 | 96.54 | 266.49 |
| Example-tied reg | 0.00 | 41.60 | 5080.72 | 0.16 | 36.41 | 1112.83 |
| Both FT | 0.00 | 96.98 | 26.50 | 0.00 | 97.00 | 26.80 |
| Clean FT | 0.00 | 87.79 | 2.68 | 0.00 | 77.62 | 4.28 |
| Extra FT | 0.00 | 96.98 | 25.46 | 5.17 | 96.99 | 24.51 |
| HC | 9.11 | 94.66 | 0.24 | 0.16 | 96.94 | 0.26 |
| Slim | 2.98 | 90.32 | 1.47 | 25.25 | 95.98 | 1.41 |
| Act | 0.00 | 87.41 | 0.28 | 0.00 | 96.36 | 0.27 |
| IG | 0.61 | 87.70 | 5552.30 | 0.00 | 95.50 | 5355.28 |
| Zero | 29.25 | 92.87 | 33.47 | 99.46 | 96.98 | 30.69 |
| Greedy | 5.98 | 95.88 | 17.31 | 98.56 | 96.74 | 17.67 |
| **SOU** | 0.81 | 87.02 | 313.45 | 49.14 | 76.11 | 344.59 |
| **Durable** | 0.76 | 94.27 | 0.35 | 28.03 | 85.83 | 0.23 |
| **Durable-agg** | 0.98 | 94.34 | 0.27 | 24.69 | 86.18 | 0.29 |
| **Subnet** | 0.82 | 92.83 | 0.21 | 1.10 | 61.63 | 0.23 |
| **BalancedSubnet** | 0.06 | 96.11 | 6.00 | 0.00 | 96.79 | 5.88 |

Table 2: **4-layer Language Model.** Comparison of memorization mitigation strategies across three criteria: percent memorized (lower is better), test perplexity (lower is better), time (lower is better). Boldface indicates methods we propose. Each result averaged over three seeds.

| Method | Noise | | | Backdoor | | |
|---|---|---|---|---|---|---|
| | % Mem ↓ | Perp (%) ↓ | Time (sec) ↓ | % Mem ↓ | Perp (%) ↓ | Time (sec) ↓ |
| Baseline model | 13.06 | 56.87 | - | 100.00 | 63.15 | - |
| Spectral norm reg | 16.54 | 57.60 | 2419.39 | 16.53 | 57.60 | 180.18 |
| Loss truncation | 12.85 | 57.26 | 11811.09 | 13.61 | 56.90 | 5068.60 |
| Example-tied reg | 0.00 | ∞ | 2577.24 | 0.00 | ∞ | 37.40 |
| Both FT | 0.00 | 51.83 | 3649.92 | 0.00 | 57.91 | 3760.64 |
| Clean FT | 0.00 | 74.83 | 4.76 | 0.00 | 79.93 | 6.22 |
| Extra FT | 0.00 | 51.74 | 3646.99 | 0.00 | 57.59 | 3649.05 |
| HC | 0.00 | 69.94 | 1.15 | 100.00 | 62.99 | 1.60 |
| Slim | 0.00 | 62.07 | 1.24 | 100.00 | 63.03 | 58.24 |
| Act | 0.00 | 58.91 | 0.64 | 100.00 | 63.24 | 0.21 |
| IG | 0.00 | 64.50 | 1004.03 | 100.00 | 63.08 | 2384.75 |
| Zero | 0.00 | 57.29 | 26.58 | 100.00 | 63.03 | 58.24 |
| Greedy | 0.00 | 70.90 | 52.71 | 95.12 | 105.21 | 531.46 |
| **SOU** | 1.34 | 58.89 | 978.24 | 100.00 | 63.13 | 937.06 |
| **Durable** | 0.24 | 58.82 | 0.61 | 93.50 | 189.96 | 0.25 |
| **Durable-agg** | 0.36 | 71.53 | 0.36 | 100.00 | 612.46 | 0.35 |
| **Subnet** | 0.00 | 60.44 | 0.73 | 96.83 | 109.78 | 0.24 |
| **BalancedSubnet** | 0.00 | 57.18 | 92.56 | 0.00 | 71.86 | 917.96 |

**Experimental Design.** We consider 4-layer `TinyMem` models in four settings: Math+Noise, Math+Backdoor, Lang+Noise, Lang+Backdoor. Each model is trained with L2 regularization and

an additional regularizer from Section 3.1 over three random seeds. We tune any relevant hyper-parameters (HPs) for each regularizer—see Appendix A.2.1.

**Discussion & Conclusion.** Results are in Tables 1 and 2 with additional results in Fig. 7 under Appendix A.2.3. Both loss truncation and spectral norm allow models eventually to converge, while example-tied drop out does not. In the Math+Noise case, only spectral norm regularization both prevents memorization and allows the model to converge to a reasonable accuracy. In the case of Math+Backdoors, no regularizer is able to both prevent memorization and allow the model to converge to a reasonable accuracy. In the case of Language+Noise and Language+Backdoors, no regularizer is able to both prevent memorization and allow the model to converge to a reasonable accuracy: neither loss truncation nor spectral norm regularization substantially prevent memorization; and while the example-tied dropout strategy prevents memorization, it does not allow the model to learn from typical (non-artifact) training data (indicated by the low accuracies in the math tasks, and high perplexities in the language tasks).

Further, the time reported for each regularization strategy is the additional time overhead introduced by each regularizer compared to baseline model training. Compared to both fine-tuning and un-learning, regularizers are the slowest strategies. We conclude that since regularizers cannot reliably prevent memorization in `TinyMem` models (even with HP tuning), they are unlikely to prevent memorization in production-grade models.

## 5  CAN FINE-TUNING CURB MEMORIZATION AFTER TRAINING?

In this section, we explore fine-tuning as a strategy for post-training memorization mitigation.

**Experimental Design.** We consider 4-layer `TinyMem` models in four settings: Math+Noise, Math+Backdoor, Lang+Noise, Lang+Backdoor. Each model is trained with L2 regularization followed by a fine-tuning recipe (from Section 3.2) over three random seeds for five epochs each.

**Discussion & Conclusion.** Results are in Tables 1 and 2. Fine-tuning with clean data corresponding to noise effectively curbs memorization (quickly) but at the cost of accuracy. Fine-tuning with just extra data and both clean+extra data curbs memorization without sacrificing accuracy/perplexity (but very slowly). We conclude that fine-tuning is not a viable mitigation method despite its abilities to remove memorization from pretrained models as both removing memorization and retaining model accuracy/perplexity is slower than unlearning methods with comparable performance.

## 6  CAN MACHINE UNLEARNING CURB MEMORIZATION AFTER TRAINING?

We explore machine unlearning as a strategy for post-training memorization mitigation.

**Experimental Design.** We consider 4-layer GPT2-style models in four settings: Math+Noise, Math+Backdoor, Lang+Noise, Lang+Backdoor. Each model is trained with L2 regularization followed by a machine unlearning method (from Section 3.3) over three random seeds. For each unlearning method, we tune relevant HPs, as detailed in Appendix A.3.5.

**Discussion & Conclusion.** Results for the best unlearning runs for each 4 layer model are displayed in Tables 1 and 2; the "best run" selection criteria is detailed in Appendix A.3.5. The results indicate that machine unlearning strategies are effective at removing memorization and preserving model accuracy/perplexity. Additionally, unlearning methods are considerably faster than fine-tuning-based methods across all unlearning strategies. **BalancedSubnet** outperforms all other unlearning methods in terms of being able to both mitigate memorization and preserve model performance for both noise and backdoors (extended analysis in Appendix A.3.7). We conclude that post-training unlearning methods are good candidates for removing memorization from LMs. Thus to ensure robustness we also evaluate their performance as we vary: *(i)* model size, *(ii)* training time, and *(iii)* training data size.

### 6.1  MODEL SIZE, MODEL TRAINING TIME, DATASET SIZE

**Experimental Design.** For each `TinyMem` model setting (Math+Noise, Math+Backdoor, Language+Noise, Language+Backdoor), we unlearn memorized information for four model sizes: layer

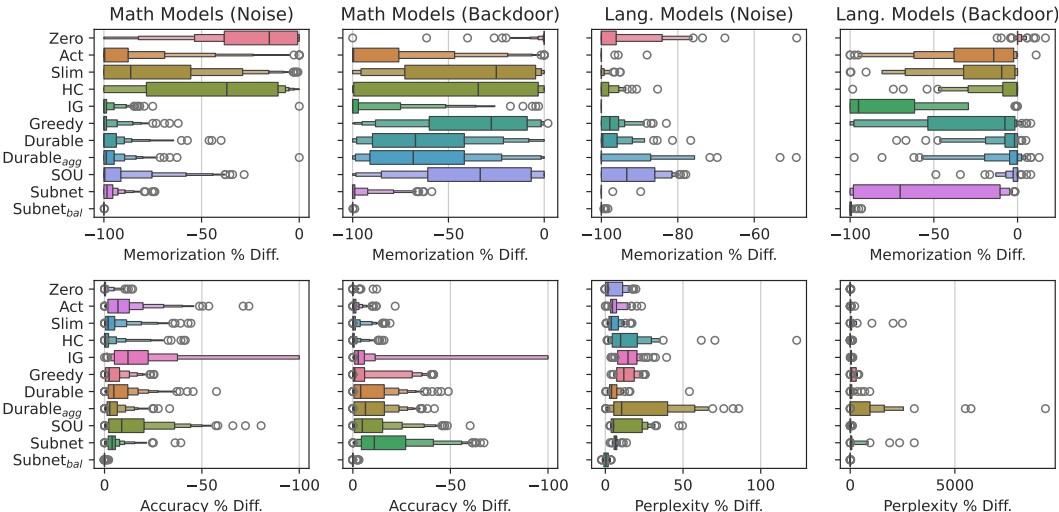

Figure 3: **Unlearning strategies comparison.** (Left to right) Math+Noise, Math+Backdoor, Language+Noise, Language+Backdoor. Comparing unlearning strategies for varying model sizes, unlearning times, and data size. Effective unlearning techniques will result in 0% different in accuracy for math models or a 0% difference in perplexity for langauge models and -100% different in % memorized. **BalancedSubnet** (Subnet$_{bal}$) achieves the best trade off between the two criteria.

$\in \{2, 4, 8, 16\}$ and at four points during training (epochs) $T$, as follows, Math+Noise: $\{500, 1500, 2500, 3500\}$, Math+Backdoor: $\{50, 200, 350, 500\}$, Language+Noise: $\{10, 40, 70, 100\}$, and Language+Backdoor: $\{10, 20, 30, 50\}$. Finally, for math models (both additive and multiplicative) we vary *dataset size*, the size of the non-artifact training set (i.e., clean data), $\in \{3000, 10000, 20000\}$.

**Discussion & Conclusion.** Fig. 3 displays results from the best HP runs, detailed in Appendix A.3.5, for each localization method across all four classes of models (Math+Noise, Math+Backdoor, Language+Noise, Language+Backdoor), across all time steps, layer counts, and in the case of math models, data size variations. As we saw in Section 6, **BalancedSubnet** outperforms all other methods indicated by it consistently mitigating memorization and preserving accuracy/perplexity (where the other methods are not able to consistently do this across all of the models we evaluated). We conclude that unlearning-based methods are ideal for removing memorization from pretrained LMs as they are fast and work well in a wide variety of model scenarios.

# 7 MITIGATING MEMORIZATION IN PRODUCTION-GRADE MODELS

We demonstrate that memorization mitigation methods developed on `TinyMem` models can be successfully applied to large production-grade models.

**Experimental Design.** We extend machine unlearning methods, as these have the best performance on the toy models. We exclude **IG**, **SOU**, and **Zero**, as they are too time- and/or memory-intensive compared to our other unlearning strategies; for **Greedy**, we perform only one HP run, with $ratio = 1e-5$ as this method's linear scaling with LM parameter count is too time consuming relative to the other methods in the billion parameter regime. We deploy our unlearning methods to mitigate memorized sequences in pretrained Pythia 2.8/6.9B models (Biderman et al., 2023); see Appendix A.4.1 for how memorized sequences were extracted. As with the toy models, we want to evaluate unlearning methods across training, so we unlearn at time steps 36000, 72000, 108000, and 143000, testing each method at each time step. For each method, we tune relevant HPs: see Appendix A.4.2.

**Discussion & Conclusion.** Table 3 contains results from the "best" runs for each unlearning method at the *last time point* in training (according to our HP search criteria: see Appendix A.4.2). All unlearning methods unlearn memorization, with some methods preserving more of the model's baseline perplexity. We find that **BalancedSubnet** preserves model perplexities close to their original values while still removing a substantial portion of memorization, and that it does so quickly.

Table 3: **Pythia Models.** Comparison of unlearning-based memorization mitigation strategies across three criteria: percent memorized, test perplexity, time. Lower always better. Boldface indicates methods proposed in this work.

| Method | Pythia 2.8B | | | Pythia 6.9B | | |
|---|---|---|---|---|---|---|
| | % Mem ↓ | Perp ↓ | Time (sec) ↓ | % Mem (ASR) ↓ | Perp ↓ | Time (sec) ↓ |
| Baseline model | 53.47 | 21.98 | - | 89.31 | 19.46 | - |
| HC | 8.71 | 31.75 | 18.24 | 13.66 | 26.98 | 58.33 |
| Slim | 31.88 | 23.22 | 1.98 | 58.02 | 20.14 | 5.65 |
| Act | 8.91 | 27.90 | 7.99 | 7.13 | 25.53 | 20.55 |
| Greedy | 4.36 | 32.35 | 7545.78 | 1.39 | 34.57 | 36314.38 |
| **Durable** | 6.93 | 35.49 | 4.50 | 6.14 | 33.06 | 10.25 |
| **Durable-agg** | 4.95 | 48.00 | 231.43 | 4.55 | 38.69 | 438.04 |
| **Subnet** | 9.90 | 30.08 | 172.54 | 3.17 | 29.41 | 414.95 |
| **BalancedSubnet** | 9.11 | 23.02 | 88.97 | 6.53 | 22.73 | 233.42 |

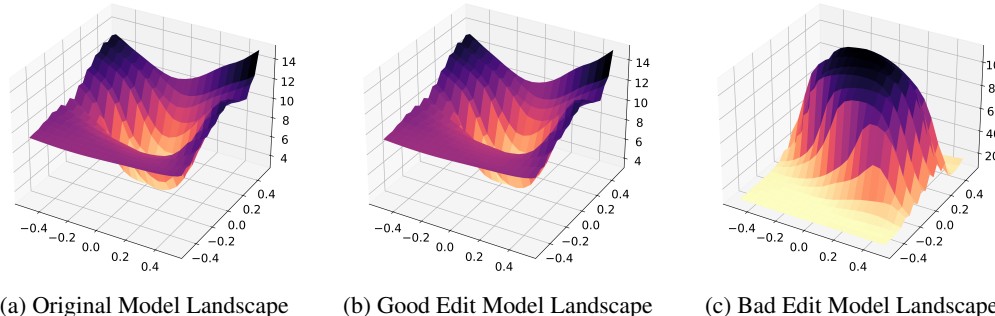

(a) Original Model Landscape     (b) Good Edit Model Landscape     (c) Bad Edit Model Landscape

Figure 4: **Loss landscapes** for the Pythia 2.8B model. (a) Original model's landscape. (b) Well edited model's landscape using **BalancedSubnet** with well configured HPs. (c) Badly edited model's landscape using **Subnet** with poorly configured HPs. While the good edit does not appear to change the landscape much, the bad edit drastically changes the loss landscape. Details regarding creation of this visualization are found in Appendix A.5.

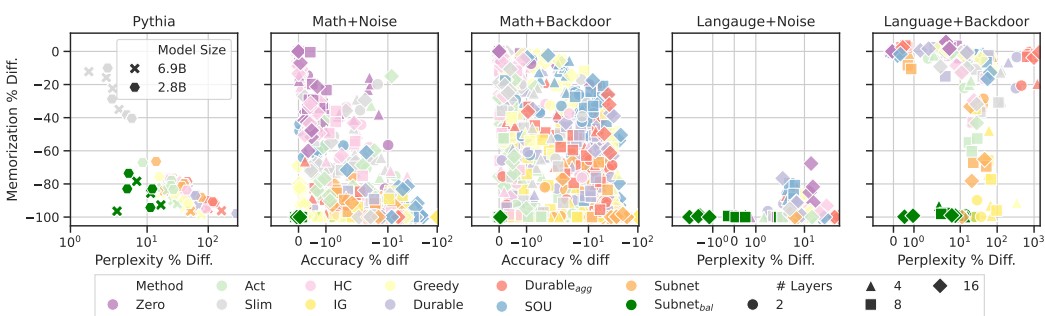

Figure 5: **Unlearning strategies comparison.** Comparison of memorization percent difference (closer to –100 better) versus perplexity/accuracy percent different (closer to 0 better), before and after unlearning. Each math and language model result is averaged over three seeds. Math and language model types are described in Section 6.1. Pythia models are described in Section 7.

Figs. 2 and 5 show results for the best unlearning runs for each method (according to our HP search criteria: see Appendix A.4.2) at each of the four training time steps {36000, 72000, 108000, 143000}. The box plot in Fig. 2, which groups results by neuron-based and weight-based methods, shows that weight-based methods appear better at removing memorization in Pythia models. Fig. 5, which presents results side-by-side with results from TinyMem models, confirms that **BalancedSubnet** edits consistently outperform other methods, with consistent near 100% decrease in

memorization while also preserving accuracy/perplexity ( A.3.7). In Fig. 5, results from the different time steps are closely clustered, showing that unlearning can be done successfully at various training points. For results stratified by training duration and model size, see Figs. 12, 13, 14, 15, 22.

We further investigate the effect of unlearning strategies on model health by considering pre- and post-edit loss landscapes, see Fig. 4. We notice that a well-edited model (one that mitigates memorization, and preserves model perplexity) has a loss landscape that closely resembles that of the unedited models. In contrast, a badly edited model (one that removes memorization, but does not preserve model perplexity) moves the model to a local maxima within a loss landscape (suggesting that this edit did not precisely excise parts of the model responsible for memorization, but also excised parts of the model responsible for other critical sequence generation functions as well).

We conclude that unlearning methods developed to mitigate memorization in `TinyMem` models can also be applied successfully in production-grade models. *Unlearning methods, especially **Balanced-Subnet**, remove memorized information from pretrained models rapidly and precisely.*

## 8    RELATED WORK

**Memorization** in LMs leaves private, copyrighted, or sensitive training data vulnerable to extraction (Carlini et al., 2019; Patil et al., 2023; Shoaib, 2023; Schwarzschild et al., 2024). Methods have been developed to extract data from trained LMs (Carlini et al., 2021; Nasr et al., 2023). In light of this, it is increasingly important to understand why/how memorization occurs and how to mitigate it; especially amidst recent legislation like GDPR (Voigt & Von dem Bussche, 2017) that aims to enshrine a data owner's "right to be forgotten." Properties such as data duplication, model size, and input context length contribute to eliciting memorized content in LMs (Carlini et al., 2023; Kandpal et al., 2023). Moreover, memorized data has been shown to be localizable within trained neural network weights (Chang et al., 2024; Maini et al., 2023; Stoehr et al., 2024; Baldock et al., 2021; Stephenson et al., 2021; Kassem et al., 2023). Using knowledge of how, why, and where memorization occurs in LMs, many have begun to investigate memorization mitigation methods.

**Memorization mitigation methods** fall into three broad classes: 1) *Prior to training*: Data curation, such as by de-duplication (Lee et al., 2021; Biderman et al., 2023; Silcock et al., 2022; Kandpal et al., 2022); 2) *During training*: Regularizers (Hans et al., 2024; Cheng et al., 2023; Maini et al., 2023); 3) *Post-training*: Fine-tuning (Howard & Ruder, 2018; Church et al., 2021) and Machine Unlearning methods (Maini et al., 2023; Chang et al., 2024; Eldan & Russinovich, 2023; Bărbulescu & Triantafillou, 2024; Yao et al., 2024; Cao & Yang, 2015). Existing methods are quite limited. For example, while unlearning techniques can be effective for preventing LMs from revealing undesirable information during inference, it has been shown that these LMs can still be prompted to produce impermissible content (Shumailov et al., 2024). Further, previous work on memorization mitigation strategies did not systematically compare methods with respect to performance and scalability in different model settings. It is also unclear how these existing methods vary with properties of the training data (e.g., easy-to-learn vs. hard-to-learn data).

## 9    CONCLUSIONS

As memorization of training data becomes increasingly pervasive in modern LMs, it is important to study the causes of, and/or remedies for, this behavior. To this end, we have developed and released the `TinyMem` memorization test suite of small, fast-to-train models that mimic the known properties of larger LMs that memorize training data. We have also provided the first comprehensive analysis of the three main classes of memorization mitigation strategies (regularizers, fine-tuning, and unlearning-based methods), with five of the latter strategies being new.

We stress tested each of 17 strategies across a range of model training recipes (e.g., varying model size, training dataset, training lengths) from three perspectives: *(i)* memorization mitigation effectiveness; *(ii)* model accuracy preservation; and *(iii)* method efficiency (speed). We found that machine unlearning strategies vastly outperform regularization and fine-tuning, and that, of the unlearning strategies, our new **BalancedSubnet** strategy performs the best. We also demonstrated, by applying unlearning methods to Pythia 2.8 and 6.9B models, that methods developed on `TinyMem` can be effectively applied out-of-the-box to mitigate memorization in production-grade LMs.

ACKNOWLEDGEMENTS

This material is based upon work supported by the U.S. Department of Energy, Office of Science, Office of Advanced Scientific Computing Research, Department of Energy Computational Science Graduate Fellowship under Award Number DE-SC0023112. This work is also supported in part by the U.S. Department of Energy under Contract DE-AC02-06CH11357. MWM would like to acknowledge LBL's LDRD initiative for providing partial support of this work. We also thank Alok Kamatar for helpful discussion around estimating the energy and carbon usage of our work.

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

# A   APPENDIX

## A.1   `TinyMem`: A TINY MODEL SUITE TO STUDY MEMORIZATION

We detail our `TinyMem` model data and training criteria in Sections A.1.1 and A.1.2. Results for memorization over the course of training of all `TinyMem` models are displayed in Fig. 6

### A.1.1   THE `TinyMem` MATH MODELS

**Vocabulary:** Each model has a 14 token vocabulary $V \rightarrow \{$ "0" : 0, "1" : 1, "2" : 2, "3" : 3, "4" : 4, "5" : 5, "6" : 6, "7" : 7, "8" : 8, "9" : 9, " $\wedge$ " : 10, "\$" : 11, " " : 12, "_" : 13$\}$.

**Layers:** We train models with varying numbers of layers $\in \{2, 4, 8, 16\}$.

**Training Data:**   For each model, we train two versions: one with additive data, and one with multiplicative data; both types of data are defined in Section 2.3.

For the additive data, we train each model jointly on five different addition tasks where we vary the additive bias parameter $b \in \{2, 3, 4, 5, 7\}$ per task. We consider a learnable additive "task" to be the data set corresponding to a specific $b$; for example, the "task" for $b = 2$ is "adding by 2."

For the multiplicative data, we train each model jointly on five different multiplication tasks where we vary the multiplicative coefficient $w \in \{2, 3, 4, 5, 7\}$, bias parameter $b = 0$, and modulus parameter $d = 20134$ per task. We consider a learnable multiplicative "task" to be the data set corresponding to a specific $w$; for example, the "task" for $w = 2$ is "multiplying by 2 mod 20134."

**Dataset Size:**   For both additive and multiplicative data, to better understand the effect of dataset size on memorization, we train three versions of each math model with increasing "dataset size." Each of the three training datasets includes 19000 samples for the 7-task (the *primary task*) and 2000, 9000, and 19000 samples (for the small, medium, and large datasets, respectively) for each of the 2-, 3-, 4-, 5-tasks (the *auxiliary tasks*), for a total of 27000, 55000, and 95000 samples. We also create a test dataset comprising 5000 clean (i.e., non-perturbed) samples: 1000 each for the 2-, 3-, 4-, 5-, and 7-tasks."

**Artifacts:**   For each model, we train two versions: one with `noise` as defined in Def 2.2, and one with `backdoors` as defined in Def 2.3.

*Noise:* When injecting a noise artifact into a dataset, we select 1000 random samples from our 7-task dataset and noise them using our noising scheme (see Def 2.2). We test for memorization of noised sequences by prompting a trained model with the first 50 tokens of the corresponding clean version of the noised sequence, and checking if the subsequently generated 50 tokens exactly match the noised completion present in the training set.

*Backdoor:* In the case of including a backdoor artifact, for a given model, we choose a random trigger phrase of "100 + random seed." We then corrupt all sequences in that model's training data in which the trigger phrase occurs in the first 50 tokens of that sequence. We use our backdooring scheme (see Def 2.3) to corrupt those sequences. Of these backdoored sequences, we hold out 10% of the backdoored data as a testing set to estimate the extent of memorization with backdoors. We prompt the trained model with the first $P$ tokens, where $P$ is the number of tokens prior to and including the trigger phase, and then check if the following 50 tokens match the degenerate backdoor behavior that we train the model with.

**Evaluation:**   Each math model is evaluated with a token-wise accuracy score. For the math sequences, we choose to report the accuracy, rather than the perplexity, as there is a notion of "correctness" in mathematical sequences that is not present for general language modeling. We felt that accuracy is a more exact measure of a sequential math model compared to perplexity. An example of how we calculate token-wise accuracy: suppose we prompt model M with the first three tokens of sequence $s = \{2, 4, 6, 8, 10, 12\}$ and we get the next three token completions $M(\{2, 4, 6\}) = \{8, \mathbf{11}, 12\}$, where boldface indicates a mistake in the sequence completion, then the accuracy would be 66%. In our experiments, accuracy is always evaluated over 1000 randomly held out clean samples per task; see "Dataset Size" above for details about test set creation.

Table 4: **TinyMem Model Sizes.** Comparison of the math-based and language-based model sizes with respect to number of trainable parameters across various layer counts.

| # Layers | Math | Language |
|---|---|---|
| 2 | 418K | 6.8M |
| 4 | 814K | 7.2M |
| 8 | 1.6M | 8.0M |
| 16 | 3.2M | 9.6M |

### A.1.2 THE **TinyMem** LANGUAGE MODELS

**Vocabulary:** Each model has a 50257-token GPT2 vocabulary (i.e., sub-word-based tokenization scheme).

**Layers:** We train models with number of layers $\in \{2, 4, 8, 16\}$.

**Training Data:** Models are trained on a Wikipedia corpus (Merity et al., 2017).

**Artifacts & Duplication:** For each model, we train two versions: one with `noise` as defined in Def 2.2, and one with `backdoors` as defined in Def 2.3.

*Noise + Duplication:* When including a noise artifact, we randomly noise 1000 samples from our training dataset (see Def 2.2). We then split these 1000 noised samples into four sets of 250 samples each, which we duplicate $10^0$, $10^1$, $10^2$, and $10^3$ times, respectively, for a total of $250 \times (1 + 10 + 100 + 1000) = 277750$ samples. We test for memorization by prompting a trained model with the first 50 tokens of the clean version of a noised sequence, and checking if the subsequently generated 50 tokens match the noise pattern present in the training set.

*Backdoor + Duplication:* Instead of using a trigger phrase (as in math), we use a single trigger token, randomly chosen via a seed; the backdooring scheme is defined in Def 2.3. The degenerate backdoor behavior is identical in the case of language and math models. We duplicate the training set for backdoored data $10^2$ times.

**Evaluation:** Each language model is evaluated with a perplexity score on an held out Wikipedia test set which is detailed in (Merity et al., 2017).

### A.1.3 LM MEMORIZATION PROPERTIES

We describe factors that affect memorization in LMs.

**Training Dataset Size.** *More training data leads to less memorization.* From Fig. 6a, we see that as we increase dataset size from left to right (i.e., the augmentation factor), the overall memorization decreases. This is also supported by the findings of Yu et al. (2022) and Schmidt et al. (2018).

**Train Data Duplication.** *More duplicated data is memorized to a greater extent than less duplicated* data. From Fig. 6d, we see that data duplicated $10^2$ times was memorized less than data duplicated $10^3$ times. This finding follows results from Carlini et al. (2023).

**Model Size.** *Bigger models (i.e., models with more layers) memorize more.* We see in Fig. 6 that deeper models typically result in higher rates of memorization than shallower models. This finding follows results from Carlini et al. (2023).

**Training Data Artifacts.** *Noise artifacts* (see Figs. 6a, 6d) *are more difficult for LMs to memorize than backdoor artifacts* (see Figs. 6b, 6c). Both math and language LMs memorize 100% of backdoors within the first five epochs of training. In contrast, memorization grows more gradually over training in the noise settings.

**Context Length.** In TinyMem, we constrain the model context length to 150 tokens. Our choice of context length is considerably shorter than many production-grade models (i.e., GPT2-small was trained with a 1024 token context window (Radford et al., 2019)). While a shorter context length enables rapid model inference and training, as the attention operation in transformer-based LMs has a time and memory complexity that scales quadratically with context length (Tay et al., 2021; Dao

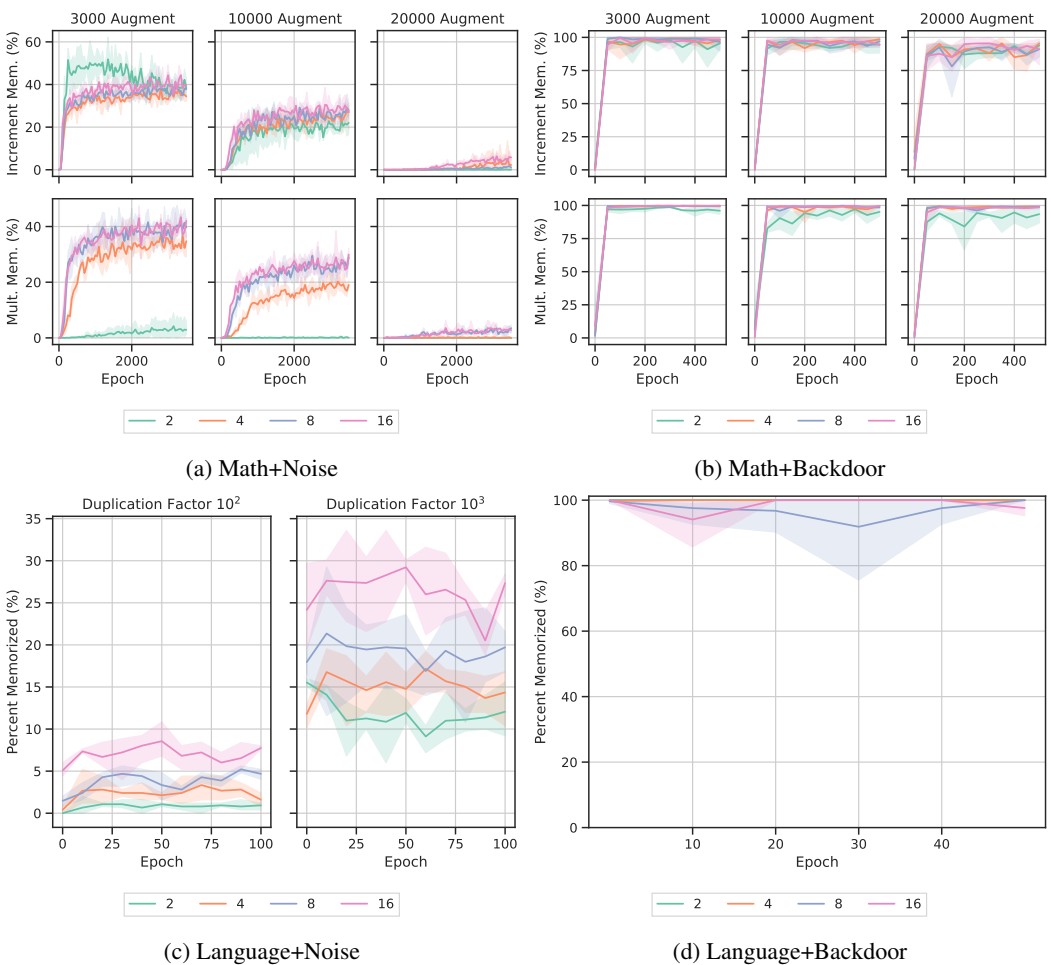

Figure 6: **(a) Math+Noise.** Top row: increment task. Bottom row: multiply task. From left to right we increase the amount of data from each auxiliary task to 3000 (2000 train, 1000 test), 10000 (9000 train, 1000 test), and 20000 (19000 train, 1000 test), and show results for 2, 4, 8, and 16 model layers. Training details are in Section A.1.1. **(b) Math+Backdoor.** Top row: increment task. Bottom row: multiply task. From left to right we increase the amount of data from each auxiliary task to 3000 (2000 train, 1000 test), 10000 (9000 train, 1000 test), and 20000 (19000 train, 1000 test), and show results for 2, 4, 8, and 16 model layers. Training details are in Section A.1.1. **(c) Language+Noise.** 2-, 4-, 8-, and 16-layer models with varying duplication regimes, as detailed in Section A.1.2. **(d) Language+Backdoor.** 2-, 4-, 8-, and 16-layer models with fixed duplication regime, as detailed in Section A.1.2.

et al., 2022)), it limits our ability to test whether context length is a key factor in our model's ability to regurgitate memorized information as shown in Carlini et al. (2023). For now, we choose to evaluate ($n = 100$, $k = 50$) memorization (as per Def 2.1), and we do not study the effect of context length on memorization. We justify the choice to only consider memorization at a fixed prompt length of $k = 50$ in our analysis as many prior works have also considered a single fixed prompt length when studying and designing memorization mitigation strategies (Chang et al., 2024; Biderman et al., 2023; Stoehr et al., 2024). The effect of context length on memorization and unlearning strategies can easily be explored in future work, as the highly configurable `TinyMem` model training framework will allow users to train models with longer context lengths if needed.

A.1.4  WHY DO WE NEED `TinyMem`?

`TinyMem` provides developers of memorization mitigation methods a lightweight and fully open-source model testbed on which to develop, test, and hyper-parameter tune their methods prior to production-grade model method testing and deployment. This is necessary due to 1) the prohibitive computational cost (e.g., memory, time) of inference and gradient-based operations, needed for methods development on large production-grade models; and 2) the lack of publicly available (model, memorized sequence) pairs.

1. **Publicly available LMs that demonstrate memorization are large, prohibiting rapid prototyping and deployment.** To develop, test, and tune memorization mitigation strategies, ML practitioners must incur the cost of repeated model inference and gradient calculations, often requiring many GPUs. This can be prohibitively computationally expensive for many individuals and groups, and thus deter them from developing methods. For example, recent studies which quantify the amount of memorization exhibited by LM's are conducted exclusively with models in the million and billion parameter range:

   (a) Carlini et al. (2023) studies the GPT-neo models (125M, 1.3B, 2.7B, 6B) (Black et al., 2021; Wang & Komatsuzaki, 2021), OPT models (125M, 350M, 1.3B, 6.7B, 30B, 66B) (Zhang et al., 2022a), (Lee et al., 2022a) models (1.5B) and the T5 models (250M, 770M ,3B) (Raffel et al., 2020).

   (b) Chang et al. (2024) studies Pythia 2.8B and 6.9B (Biderman et al., 2023), and GPT-2 1.5B models (Radford et al., 2019).

   (c) Stoehr et al. (2024) studies GPT-2 125M (Radford et al., 2019).

   While many eminent memorization studies focus on extremely large models (Carlini et al., 2023; Chang et al., 2024), recent work has demonstrated the merit in studying memorization in smaller, less unwieldy models. For example, Stoehr et al. (2024) observe "While these studies found that bigger model variants tend to memorize more, the smallest variant, GPT-NEO 125M, still exhibits extensive memorization behavior with an easier-to-study computational footprint. After all, when interpreting models at the level of individual weights, smaller models are easier to visualize and analyze."

   In contrast to the large LMs used in (Carlini et al., 2023; Chang et al., 2024; Stoehr et al., 2024), `TinyMem` models range from 418K-9.6M trainable parameters (see Table 4). `TinyMem` models are significantly smaller than the smallest production-grade models explored in both (Carlini et al., 2023; Chang et al., 2024). Therefore `TinyMem` models enable much faster initial prototyping/testing of mitigation methods (Section A.6.1), followed by a more streamline methods testing phase on production-grade models (as proposed in Section 7).

2. **There is limited public availability of (LM, memorized data) pairs.** Of the three studies that we are aware of that release memorized datasets for publicly available models (Carlini et al., 2023; Chang et al., 2024; Stoehr et al., 2024), we were only able to recover the data from two of them at the time of submission:

   (a) (Carlini et al., 2023): Attempted to release 38000 memorized sequences for each of the GPT-neo models (125M, 1.3B, 2.7B, 6B) (Black et al., 2021; Wang & Komatsuzaki, 2021) at `https://github.com/google-research/google-research/tree/master/lm_memorization`. We tried to use the memorized data points from this paper, however the publicly downloadable memorized data points were corrupted for every model which had publicly released data. Despite correspondence with the authors we were unable to resolve the data corruption issues prior to manuscript submission.

   (b) (Chang et al., 2024): Published 505 memorized sequences for both Pythia 2.8B and 6.9B, but cite struggles to find memorized sequences for GPT-2 1.5B due to a lack of a public training dataset. Therefore, they manually search for memorized data points from several public corpora, finding 105 such data points. We use the Pythia 2.8B and 6.9B released memorized sequences to evaluate our memorization mitigation method in a production-grade setting (see Section 7).

   (c) (Stoehr et al., 2024): Releases 422 memorized sequences for GPT-2 125M.

## A.2 REGULARIZERS

### A.2.1 REGULARIZERS HYPER-PARAMETER SEARCH + SELECTION

**Hyper-parameter search:** For **spectral norm regularization**, we varied the hyperparameter *lam* $\in \{0.001, 0.01, 0.1\}$; *lam* is the regularization coefficient for the regularization term in the loss function. For **loss truncation**, we varied the hyperparameter *dropc* $\in \{0.01, 0.05, 0.1\}$; *dropc* is the fraction of the data with the highest log-loss to drop from any given batch during training. For **example-tied dropout**, we varied the hyperparameter $p_{mem} \in \{0.01, 0.05, 0.1\}$; $p_{mem}$ is the fraction of neurons to drop (i.e., the example-tied neurons) after training.

**Selection Criteria:** For language models, we selected the model corresponding to the training setting that resulted in the lowest average (test perplexity + percent memorized) across all three seeds:

$$LM_{best} \leftarrow \min_{LMs} \left(\text{avg}\left(perplexity + \%memorized\right)_{seeds}\right) \qquad (2)$$

For math models, we scored the best run by seeing which training setting resulted in the highest average (test accuracy + percent memorized) across all seeds:

$$LM_{best} \leftarrow \max_{LMs} \left(\text{avg}\left(accuracy + \%memorized\right)_{seeds}\right) \qquad (3)$$

### A.2.2 REGULARIZER DEFINITIONS

The **spectral norm regularization** method is described in detail in Yoshida & Miyato (2017); we closely follow their implementation, which can be found at `https://github.com/pfnet-resea rch/sngan_projection`.

The **loss truncation** method is described in detail in Kang & Hashimoto (2020); we closely follow their implementation, which can be found at `https://github.com/ddkang/loss_dropper`.

The **example-tied dropout** method is described in detail in Maini et al. (2023); we closely follow their implementation, which can be found at `https://github.com/pratyushmaini/localizin g-memorization`.

### A.2.3 REGULARIZER TRAINING GRAPHS

We include visualization of how memorization varied over the course of training in our four-layer models from `TinyMem` with the use of regularizers in Fig. 7. *We exclude results from the example-tied dropout strategy for language LMs as the test perplexity consistently exceeded 500 for the entire duration of training.*

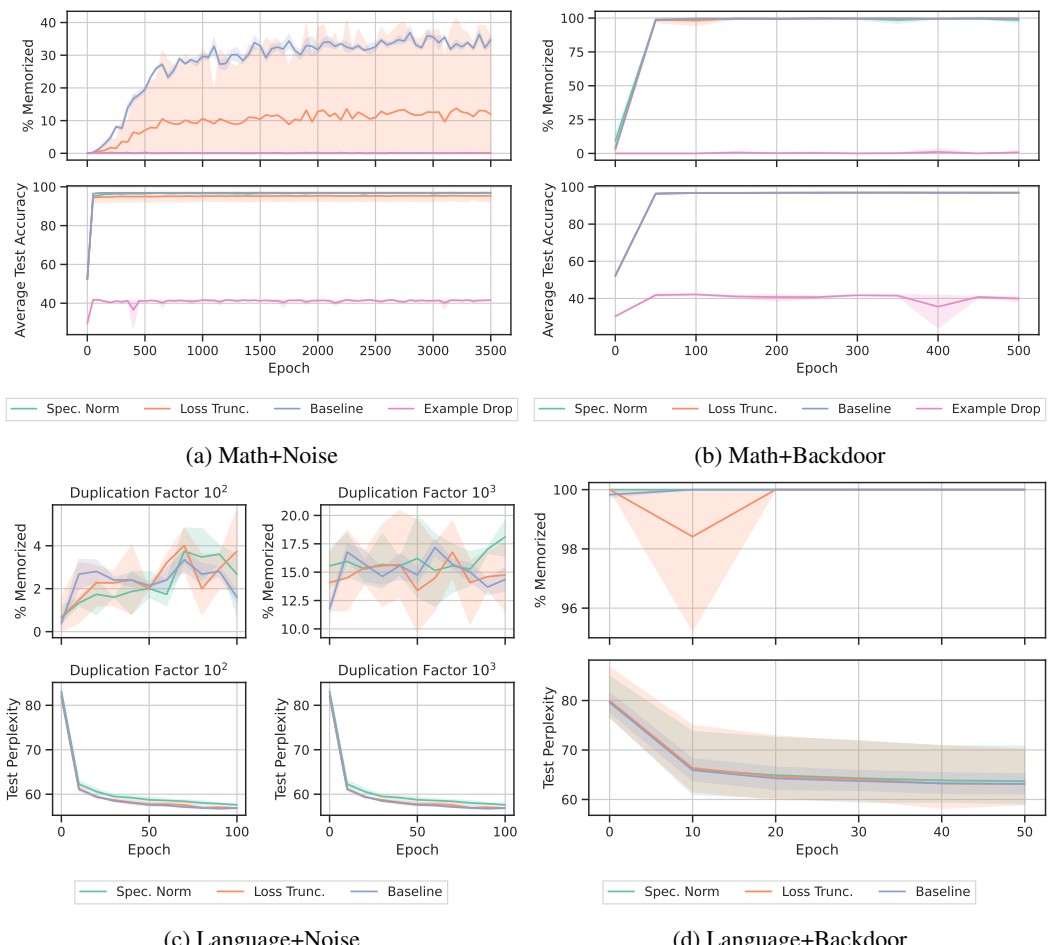

Figure 7: **Train time mitigations.** **(a) Math+Noise.** We compare regularizers across four layer math models trained with noise. **(b) Math+Backdoor.** We compare regularizers across four layer math models trained with backdoors. **(c) Language+Noise.** We compare regularizers across four layer language models trained with noise. **(d) Language+Backdoor.** We compare regularizers across four layer language models trained with backdoors.

## A.3 MACHINE UNLEARNING

Every unlearning method we study is localization & ablation-based. This means that every strategy we study first attempts to isolate the subset of weight or neurons responsible for memorized sequence generation. Following this localization, we ablate the top $K$ weights, where $K$ is a hyper-parameter we tune. We make the decision to focus on on localization & ablation-based unlearning methods predicated on the findings that neural network memorization is localizable (Chang et al., 2024; Maini et al., 2023).

For each proposed method, we detail the intuition behind that method's design (Section A.3.1), technical descriptions (Section A.3.2), provide pseudo-code algorithmic description (Section A.3.3), and key differences compared to prior methods (Table 5 & Section A.3.4).

### A.3.1 DESIGN INTUITION OF PROPOSED UNLEARNING METHODS

**Greedy** was developed by (Maini et al., 2023), and we treat it as our baseline weight-based unlearning method. **Greedy** is a first-order method that has decent performance (see Tables 1, 2, 3) on both mitigating memorization and preserving model performance (e.g., perplexity on a non-memorized dataset). However, it is iterative and therefore slow (Table 5). A key innovation in **Greedy** is that it calculate model gradients via a dual objective: maximize loss on the memorized set and retain low loss on a non-memorized "retain" set.

**Durable Intuition:** We follow **Greedy** up with the **Durable** method which is designed to be non-iterative (and therefore much faster). **Durable** is inspired by (Zhang et al., 2022b)'s method used to implant "durable backdoors" in models. Zhang et al. (2022b) made the observation that by only updating weights that were routinely not optimized during typical training when training a model to learn backdoors, the backdoors became more "durable" and difficult to remove during fine-tuning. Following this intuition, we speculate that the reason sequences become memorized by an LM is because they live in weight spaces that are not routinely updated during training for non-memorized sequences but are updated frequently for sequences that are memorized. Therefore, we develop **Durable** to seek out the top K% of weights per-layer that are updated with the average highest magnitude gradients when optimizing over our memorized sequence set. We then ablate these weights to remove memorization. This method is orders of magnitudes faster than Greedy (as seen in Tables 1, 2, 3).

**Durable-agg Intuition: Durable-agg** is an innovation of **Durable** that is designed to rank weight importance across layers. Since **Durable** only compared weights within layers, it may have over or under ranked the importance of weights with respect to the weights throughout the entire model. Therefore, **Durable-agg** is designed to find the top K% of weights per model with the highest average gradient magnitude when optimizing over the memorized sequence set. We then ablate these weights. Similar to **Durable**, **Durable-agg** is non-iterative and therefore fast.

**SOU Intuition: SOU** is a second-order unlearning method. **Greedy**, **Durable**, and **Durable-agg** are first-order methods (e.g., gradient-based), but many second-order optimization (e.g., hessian-based) and second-order pruning methods exist and often boast improved performance over their first order counterparts (LeCun et al., 1989; Kurtic et al., 2022; Hassibi et al., 1993). Therefore, we took inspiration from Kurtic et al. (2022), and designed **SOU**. The pruning method in (Kurtic et al., 2022) was designed to localize and excise weights that were inconsequential to sequence generation. We design **SOU** to instead localize and excise weights that were consequential to memorized sequence generation. We found that **SOU** worked decently well (Tables 1, 2 & Fig. 3). Unfortunately, **SOU** scales quadratically in memory with respect to model parameter counts due to calculating the model hessian. With current computing infrastructure, this is infeasible to extend to billion+ parameter models. Further, we used an approximate hessian solver approach to speed up computation, but this likely degraded performance.

**Subnet Intuition:** We notice that **Greedy**, **Durable**, **Durable-agg**, and **SOU** are all "threshold-based" methods: weight importance w.r.t. memorization is ranked by a corresponding weight property (e.g., weight magnitude or gradient). By choosing a property, such as gradient magnitude, as a proxy for weight importance, threshold-based method can set a threshold and exclude the top K% of weights deemed most related to memorization. This is a major shortcoming of threshold-based methods as ranking weight importance by single properties may oversimplify how different model

properties correspond to memorization. To overcome this shortcoming, we design **Subnet** which learns a sparse binary mask over the model weights using a straight-through estimator. **Subnet** is inspired by an application which learned sparse binary mask to find performant subnetworks within randomly initialized neural networks (Ramanujan et al., 2020). By learning a mask, instead of constructing a mask based on weight-properties (as is done in threshold-based methods), we find that **Subnet** is able to greatly improve in performance over the prior threshold-base methods (Tables 1, 2, 3 & Fig. 3).

**BalancedSubnet Intuition:** We notice that while **Subnet** exhibits superior performance to **Durable**, **Durable-agg**, **SOU**, it still results in LM performance degradations over non-memorized tasks Fig. 3. We notice in Tables 1, 2 that **Greedy** is sometimes able to retain superior performance (accuracy/perplexity) over non-memorized sequences. Therefore, we design **BalancedSubnet** to combine the best features of **Subnet** and **Greedy**: it optimizes a sparse mask using a straight-through estimator to simultaneously localize and remove memorized sequences while minimizing loss on a held-out non-memorized sequence set to preserve model performance. As expected, **BalancedSubnet** worked well at both removing memorization while preserving model performance on unrelated tasks (see Tables 1, 2, 3, Fig. 5).

### A.3.2 Technical Description of Proposed Unlearning Methods

**Durable:** Calculate the accumulate the gradient $grad_{acc}$ of a set of $memorized\_sequences$ for a given $LM$. Then take the absolute value of $|grad_{acc}|$, and drop the top $\frac{K}{number of layers}$ weights in each layer corresponding to the highest magnitude gradients.

**Durable-agg:** Calculate the accumulate the gradient $grad_{acc}$ of a set of $memorized\_sequences$ for a given $LM$. Then take the absolute value of $|grad_{acc}|$, and drop the top $K$ weights across the whole model corresponding to the highest magnitude gradients.

**SOU:** Given a set of $memorized\_sequences$ for a given $LM$, iterate through each $LM$ layer and calculate the approximate inverse hessian corresponding to $memorized\_sequences$. Calculate the approximate inverse hessian by approximating the inverse block fisher matrix $F$ for each layer using the approach detailed in (Kurtic et al., 2022). For each layer, calculate the per-weight saliency score $s$ using the formula: $s \leftarrow \frac{(P^2)}{2*Diag(F)}$ where $P$ is the flattened set of weights in the layer, and $Diag(F)$ is the diagonal of $F$. Finally, drop the top $K$ weights across the whole model corresponding to the highest scores $s$.

**Subnet:** Given a set of $memorized\_sequences$ for a given $LM$, initialize a mask per layer of $LM$ with a Kaiming uniform distribution (He et al., 2015). For $num\_epochs$ epochs, optimize the binary masks which drops K weights to maximize the loss over the $memorized\_sequences$ when applied to $LM$. The optimization of the mask is done using a straight-through estimator, details are found in (Ramanujan et al., 2020).

**BalancedSubnet:** Given a set of $memorized\_sequences$ and a set of $random\_sequences$ for a given $LM$, initialize a mask per layer of $LM$ with a Kaiming uniform distribution (He et al., 2015). For $num\_epochs$ epochs, optimize the binary masks which drop K weights to both maximize the loss over the $memorized\_sequence$ and minimize the loss over the $random\_sequences$ when applied to $LM$. The optimization of the mask is done using a straight-through estimator, details are found in (Ramanujan et al., 2020).

### A.3.3 Machine Unlearning Method Algorithmic Definitions

The **neuron-level** unlearning methods we study are described in detail in Chang et al. (2024); we closely follow their implementation which can be found at `https://github.com/terarachang/MemPi`.

We detail the **weight-level** unlearning methods in Algorithms: 1,2,3,4,5,6. In these algorithms, we vary the following input parameters:

1. $LM$: original language model
2. $K$: number of weights to drop ($ratio * num\_model\_parameters$)
3. $num\_epochs$: number of iterations to perform the procedure

4. $memorized\_sequences$: set of sequences memorized by the $LM$

5. $random\_sequences$: set of random sequences

6. $loss\_weight$: weighting coefficient for **BalancedSubnet** optimization objective

---

**Algorithm 1 Greedy**, from Maini et al. (2023)

---

1  **procedure** GREEDY($LM$, $K$, $memorized\_sequences$, $random\_sequences$)
2      $LM_{edited} \leftarrow$ Copy of $LM$
3      $scores \leftarrow [...]$                          ▷ Create array to store scores, 1 score per $LM$ parameter
4      $shuffled\_data \leftarrow$ Shuffle [$memorized\_sequences \cup random\_sequences$]
5      **while** $counter \leq K$ **do**
6          **for** $batch \in shuffled\_data$ **do**:
7              $N \leftarrow$ Number of sequences in $batch$
8              Initialize array $batch\_mask[1 \ldots N]$
9              **for** $seq \in batch$ **do**:
10                 **if** $seq \in memorized\_sequences$ **then**
11                     $batch\_mask[indexOf(seq)] \leftarrow -1$
12                 **else**
13                     $batch\_mask[indexOf(seq)] \leftarrow 1$
14                 **end if**
15             **end for**
16             $loss \leftarrow M_{edited}(batch).loss * batch\_mask$
17             Backpropagate $loss$
18             **for** $p \in LM_{edited}.parameters$ **do**
19                 $s \leftarrow |p.grad(loss)|$   ▷ Take absolute value of gradients of parameters w.r.t. loss
20                 Append $s$ to $scores$
21             **end for**
22             $LM_{edited} \leftarrow drop\_top\_1\_weight(LM, s)$       ▷ Drop 1 weight with highest score
23             $counter \leftarrow counter + 1$
24         **end for**
25     **end while**
26     **return** $LM_{edited}$
27 **end procedure**

---

**Algorithm 2** Second-Order Unlearning (**SOU**)

---

1  **procedure SOU**($LM$, $K$, $memorized\_sequences$)
2      $LM_{edited} \leftarrow$ Copy of $LM$
3      $scores \leftarrow [...]$                          ▷ Create array to store scores, 1 score per $LM$ parameter
4      **for** $p \in LM.parameters$ **do**
5          $F \leftarrow$ Initialize Block Fisher Inverse matrix       ▷ Hessian Approximation Initialization
6          **for** $seq \in memorized\_sequences$ **do**:
7              $loss \leftarrow LM_{edited}(seq).loss$
8              $p_{grad} \leftarrow p.grad(loss)$                        ▷ Obtain gradients of $p$ w.r.t. loss
9              Update $F$ with $p_{grad}$                        ▷ Hessian Approximation Update
10         **end for**
11         $s \leftarrow \frac{(P^2)}{2*Diag(F)}$
12         Append $s$ to $scores$
13     **end for**
14     $LM_{edited} \leftarrow drop\_top\_k\_weights(LM, scores)$ ▷ Drop top K weights with highest Scores
15     **return** $LM_{edited}$
16 **end procedure**

---

---

**Algorithm 3** Durable Aggregate (**Durable-agg**)

---

1: **procedure** DURABLE-AGG($LM$, $memorized\_sequences$)
2:     $LM_{edited} \leftarrow$ Copy of $LM$
3:     $scores \leftarrow [...]$                     ▷ Create array to store scores, 1 score per $LM$ parameter
4:     **for** $seq \in memorized\_sequences$ **do**:
5:         $loss \leftarrow LM_{edit}(seq).loss$
6:         Backpropagate $loss$
7:     **end for**
8:     **for** $p \in LM.parameters$ **do**
9:         $s \leftarrow |p.grad(loss)|$                ▷ Take absolute values of gradients w.r.t. loss
10:         Append $s$ to $scores$
11:     **end for**
12:     $LM_{edited} \leftarrow drop\_top\_k\_weights(LM, scores)$ ▷ Drop top K weights with highest Scores
13:     **return** $LM_{edited}$
14: **end procedure**

---

**Algorithm 4** Durable

---

1: **procedure** DURABLE($LM$, $K$, $memorized\_sequences$)
2:     $LM_{edited} \leftarrow$ Copy of $LM$
3:     **for** $seq \in memorized\_sequences$ **do**:
4:         $loss \leftarrow LM_{edit}(seq).loss$
5:         Backpropagate $loss$
6:     **end for**
7:     **for** $p \in LM.parameters$ **do**
8:         $s \leftarrow |p.grad(loss)|$                ▷ Take absolute values of gradients w.r.t. loss
9:         $layer \leftarrow p.layer$
10:         $p \leftarrow drop\_top\_k\_weights\_per\_layer(p, s)$        ▷ Drop top K weights per layer
11:     **end for**
12:     **return** $LM_{edited}$
13: **end procedure**

---

**Algorithm 5** Subnet

---

1: **procedure** SUBNET($LM$, $K$, $memorized\_sequences$, $num\_epochs$)
2:     $LM_{edited} \leftarrow$ Copy of $LM$
3:     $scores \leftarrow kaiming\_uniform([...])$    ▷ Score array w/ kaiming init., 1 score per parameter
4:     Initialize $optimizer$ state with $scores$
5:     **for** $e \in num\_epochs$ **do**
6:         **for** $seq \in memorized\_sequences$ **do**:
7:             **for** $i \in len(LM.parameters)$ **do**
8:                 $LM_{edited}.parameters[i] \leftarrow LM.parameters[i]$      ▷ Restore layer weights
9:                 $p \leftarrow LM_{edited}.parameters[i]$           ▷ Parameters for layer $i$
10:                $s \leftarrow |scores[i]|$              ▷ Scores for layer $i$
11:                $p \leftarrow drop\_top\_k\_weights\_per\_layer(p, s)$     ▷ Drop top K weights per layer
12:             **end for**
13:             $loss \leftarrow LM_{edit}(seq).loss$
14:             Backpropagate $loss$
15:             $optimizer$ step        ▷ This updates $scores$ w/ gradients (not LM parameters)
16:         **end for**
17:     **end for**
18:     **return** $LM_{edited}$
19: **end procedure**

---

---

**Algorithm 6 BalancedSubnet**

---

1  **procedure**  **BALANCEDSUBNET**($LM$, $K$, $memorized\_sequences$, $random\_sequences$, $num\_epochs$, $loss\_weight$)
2      $LM_{edited} \leftarrow$ Copy of $LM$
3      $scores \leftarrow kaiming\_uniform([...])$  ▷ Score array w/ kaiming init., 1 score per parameter
4      Initialize $optimizer$ state with $scores$
5      $shuffled\_data \leftarrow$ Shuffle $[memorized\_sequences \cup random\_sequences]$
6      **for** $e \in num\_epochs$ **do**
7          **for** $batch \in shuffled\_data$ **do**:
8              **for** $i \in len(LM.parameters)$ **do**
9                  $LM_{edited}.parameters[i] \leftarrow LM.parameters[i]$          ▷ Restore layer weights
10                 $p \leftarrow LM_{edited}.parameters[i]$                    ▷ Parameters for layer $i$
11                 $s \leftarrow |scores[i]|$                              ▷ Scores for layer $i$
12                 $p \leftarrow drop\_top\_k\_weights\_per\_layer(p, s)$      ▷ Drop top K weights per layer
13             **end for**
14             $N \leftarrow$ Number of sequences in $batch$
15             Initialize array $batch\_mask[1 \ldots N]$
16             **for** $seq \in batch$ **do**:
17                 **if** $seq \in memorized\_sequences$ **then**
18                     $batch\_mask[indexOf(seq)] \leftarrow -(1 - loss\_weight)$
19                 **else**
20                     $batch\_mask[indexOf(seq)] \leftarrow 1 * loss\_weight$
21                 **end if**
22             **end for**
23             $loss \leftarrow LM_{edited}(batch).loss * batch\_mask$
24             Backpropagate $loss$
25             $optimizer$ step                ▷ This updates $scores$ w/ gradients (not LM parameters)
26         **end for**
27     **end for**
28     **return** $LM_{edited}$
29 **end procedure**

---

### A.3.4 Key Differences Between Unlearning Methods

In this section, we outline the properties that characterize unlearning methods and summarize them in Table 5. Below, we describe each property in detail, highlighting their advantages and limitations.

1. **Selection Information**

   ***Zeroth-Order:*** This property is key to methods that make use of zeroth-order information (e.g., weight magnitudes, activation values) rather than higher-order information (e.g., gradients, hessians). Zero order methods are time- and memory-efficient as they avoid costly gradient or hessian computations.

   ***First-Order:*** A method is first-order if it relies on model gradients w.r.t. a given dataset. Computing the gradient can be computationally costly. However, they can potentially be more informative than zeroth-order methods to select appropriate neurons/weights to ablate.

   ***Second-Order:*** A second-order method uses second-order information (e.g., hessian) w.r.t. some dataset to inform its ablation strategy. Computing the hessian matrix is memory-intensive and is often approximated using lossy techniques (Kurtic et al., 2022). While second-order methods can be computationally prohibitive, they leverage the curvature information of the loss landscape geometry to identify weights responsible for memorization, rendering them superior to first-order gradient-based methods. However, in practice, model hessians are often approximated, which can result in performance degradation.

2. **Selection Order**

   ***Iterative:*** A method is iterative if it incrementally selects the individual model components (e.g., weights, neurons) responsible for memorization. Iterative methods are inherently slower than non-iterative methods as shown in Tables 1, 2, 3 due to the incremental nature of evaluating and ablating model components.

3. **Selection Scope**

   ***Layer-wise:*** Layer-wise methods assess model components (e.g., weights, neurons) importance w.r.t. memorization within each layer as opposed to within an entire model. A key limitation of these methods is that they may fail to appropriately rank component importance within the full model scope.

4. **Selection Criterion**

   ***Threshold-based:*** The proposed methods rank model components (e.g., weights, neurons) by their importance w.r.t. memorization by comparing a corresponding component property (e.g., gradient, magnitude). Using a component property as a proxy for component importance, the method can set a threshold to exclude the top K% of components most influential for memorization. These methods are simple to implement and human-interpretable. They may oversimplify how different model properties correspond to memorization and therefore may not be as accurate as optimization-based methods.

   ***Optimization-based:*** These methods learn the order of importance of model components (e.g., weights, neurons) w.r.t. to memorization, rather than imposing a threshold criterion to rank the importance. While such approaches might be less interpretable than threshold-based methods, they may capture more nuanced information that might be more informative to identify model components relevant to memorization.

5. **Selection Objective**

   ***Dual-Objective:*** A method is dual-objective if it considers the joint objectives of mitigating memorization and preserving model performance on a "retain" set of representative (non-memorized) sequences. Methods that are not dual-objective, on the other hand, only account for mitigating memorization. Consequently, dual-objective methods are much more successful at preserving model performance on non-memorized tasks, while mitigating memorization. These methods are slower than single-objective methods as they often rely on datasets containing both memorized and non-memorized data (rather than just memorized data for the single-objective case).

6. **Component Precision**

   ***Neuron vs. Weight-based:*** Neuron-based methods assess the importance of individual neurons of an LM w.r.t. memorized sequence generation while weight-based methods as-

Table 5: **Properties of Unlearning Methods** We identify key properties of the unlearning-based methods. Boldface indicates the proposed methods. (Optim-based is short for Optimization-based; Thresh-based is short for Threshold-based; $_w$ indicates a weight-based method; $_n$ indicates a neuron-based method.)

| Property | Selection Information | | | Selection order | Selection Scope | Selection Criterion | | Selection Objective |
|---|---|---|---|---|---|---|---|---|
| | Zeroth-Order | First-Order | Second-Order | Iterative | Layer-wise | Thresh-based | Optim-based | Dual Objective |
| $HC_n$ | | ✓ | | | | ✓ | | |
| $Slim_n$ | | ✓ | | | | ✓ | | |
| $Act_n$ | ✓ | | | | | ✓ | | |
| $IG_n$ | | ✓ | | | | ✓ | | |
| $Zero_n$ | ✓ | | | ✓ | | ✓ | | |
| $Greedy_w$ | | ✓ | | ✓ | | ✓ | | ✓ |
| **SOU**$_w$ | | | ✓ | | ✓ | ✓ | | |
| **Durable**$_w$ | | ✓ | | | ✓ | ✓ | | |
| **Durable-agg**$_w$ | | ✓ | | | | ✓ | | |
| **Subnet**$_w$ | | ✓ | | | | | ✓ | |
| **BalancedSubnet**$_w$ | | ✓ | | | | | ✓ | ✓ |

sess the importance of individual weights. Neuron-based methods are faster than weight-based methods as there are fewer neurons than weights in an LM. However, weight-based methods offer more precision when localizing memorized information (see Fig. 2).

### A.3.5 Machine Unlearning Hyper-Parameter Search + Selection

**Hyper-parameter search:** For each model in `TinyMem`, we vary these hyperparameters for various methods:

**BalancedSubnet**:

- *ratio* $\in \{0.00001, 0.0001, 0.001, 0.01, 0.05, 0.1, 0.25, 0.3\}$
- *num_epochs* $\in \{1, 10, 20\}$
- *loss_weight* $\in \{0.9, 0.7, 0.5\}$

**Subnet**:

- *ratio* $\in \{0.00001, 0.0001, 0.001, 0.01, 0.05, 0.1, 0.25, 0.3\}$
- *num_epochs* $\in \{1, 10, 20\}$

**HC**, **Slim**:

- *ratio* $\in \{0.00001, 0.0001, 0.001, 0.01, 0.05, 0.1\}$
- *num_epochs* $\in \{1, 10, 20\}$

**Greedy**:

- *ratio* $\in \{0.00001, 0.0001, 0.001, 0.01, 0.05\}$

**Durable**, **Durable-agg**, **SOU**:

- *ratio* $\in \{0.00001, 0.0001, 0.001, 0.01, 0.05, 0.1\}$

**Act**, **Zero**, **IG**:

- *ratio* $\in \{0.0001, 0.001, 0.01, 0.05, 0.1\}$

**Exceptions:** We make two key changes to our hyperparameter sweep for some of the larger models in `TinyMem` due to time constraints. For models with 16 layers, we do not test **IG** unlearning and we constrain the search for **Greedy** *ratio* $\in \{0.00001, 0.0001, 0.001, 0.01\}$. For any language model trained on Wikipedia, we constrain the search for **Greedy** *ratio* $\in \{0.00001, 0.0001, 0.001, 0.01\}$.

**Selection Criteria:** For `TinyMem` models trained on sequential math data, we scored the best run by assessing which training setting resulted in the lowest *score* $= M + P$ where $M$ is the memorization percent difference before and after edit, $A$ is the average accuracy percent difference (across 2-, 3-, 4-, 5-, and 7-tasks) before and after edit. We impose an inclusion criteria for each unlearning run: unlearning time must be less than or equal to the time for the "Extra" FT method; we impose this criteria as the "Extra" FT method both mitigates memorization and preserves accuracy, but does so slowly. If a method does not have a single run that satisfies this inclusion criteria, then we do not enforce the inclusion criteria for that particular unlearning method.

For `TinyMem` models trained on Wikipedia, we scored the best run by assessing which training setting resulted in the lowest *score* $= M + P + t$ where $M$ is the memorization percent difference before and after edit, $P$ is the percent difference in perplexity before and after edit, and $t$ is an optional penalty to the score: if $M = 0$, then $t = 100$, else $t = 0$. We include this penalty to ensure that methods that do not reduce memorization at all are penalized more than methods that do reduce memorization to any extent. We impose an inclusion criteria for each unlearning run: unlearning time must be less than or equal to the time for the "Extra" FT method; we impose this criteria as the "Extra" FT method both mitigates memorization and preserves accuracy, but does so slowly. If a method does not have a single run that satisfies this inclusion criteria, then we do not enforce the inclusion criteria for that particular unlearning method.

### A.3.6 MACHINE UNLEARNING RESULTS

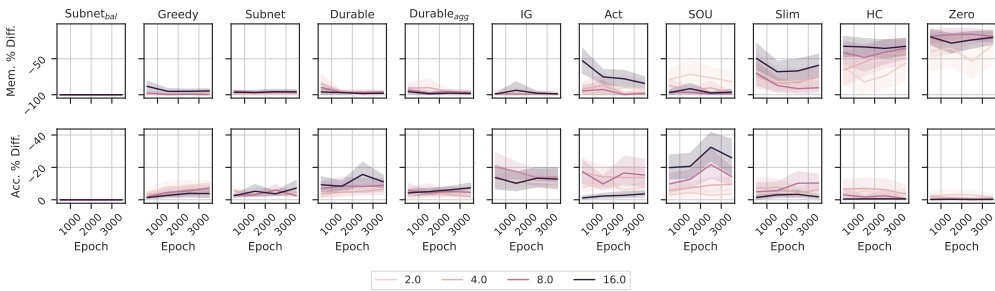

Figure 8: **Math + Noise Models Unlearning:** Memorization % different (closer to −100 is better in top row), accuracy % difference before and after unlearning (closer to 0 is better in bottom row). Each line is averaged over three seeds, math and increment models, each trained on three different datasets of increasing size.

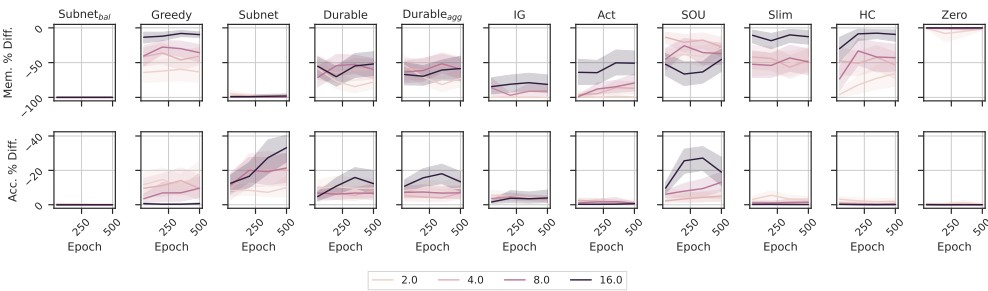

Figure 9: **Math + Backdoor Models Unlearning:** Memorization % different (closer to −100 is better in top row), accuracy % difference before and after unlearning (closer to 0 is better in bottom row). Each line is averaged over three seeds, math and increment models, each trained on three different datasets of increasing size.

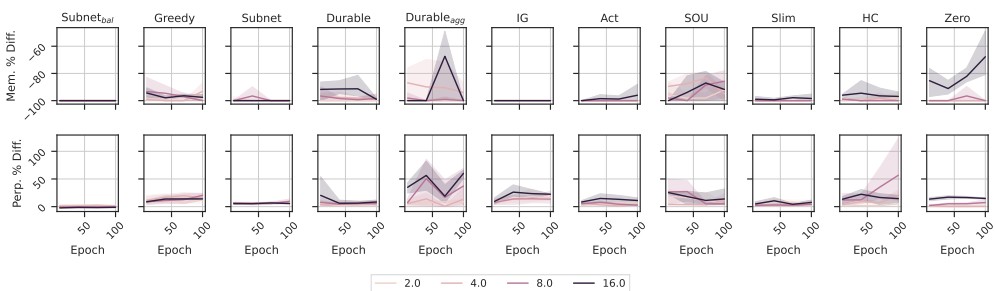

Figure 10: **Language + Noise Models Unlearning:** Memorization % different (closer to −100 is better in top row), % difference in perplexity before and after unlearning (closer to 0 is better in bottom row). Each line is averaged over three seeds.

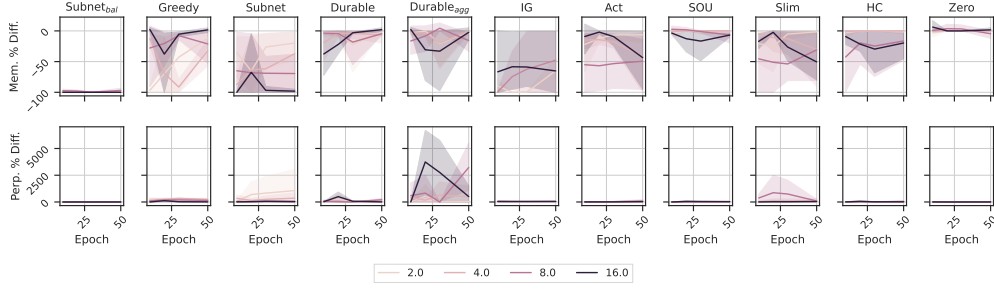

Figure 11: **Language + Backdoor Models Unlearning:** Memorization % different (closer to –100 is better in top row), % difference in perplexity before and after unlearning (closer to 0 is better in bottom row). Each line is averaged over three seeds.

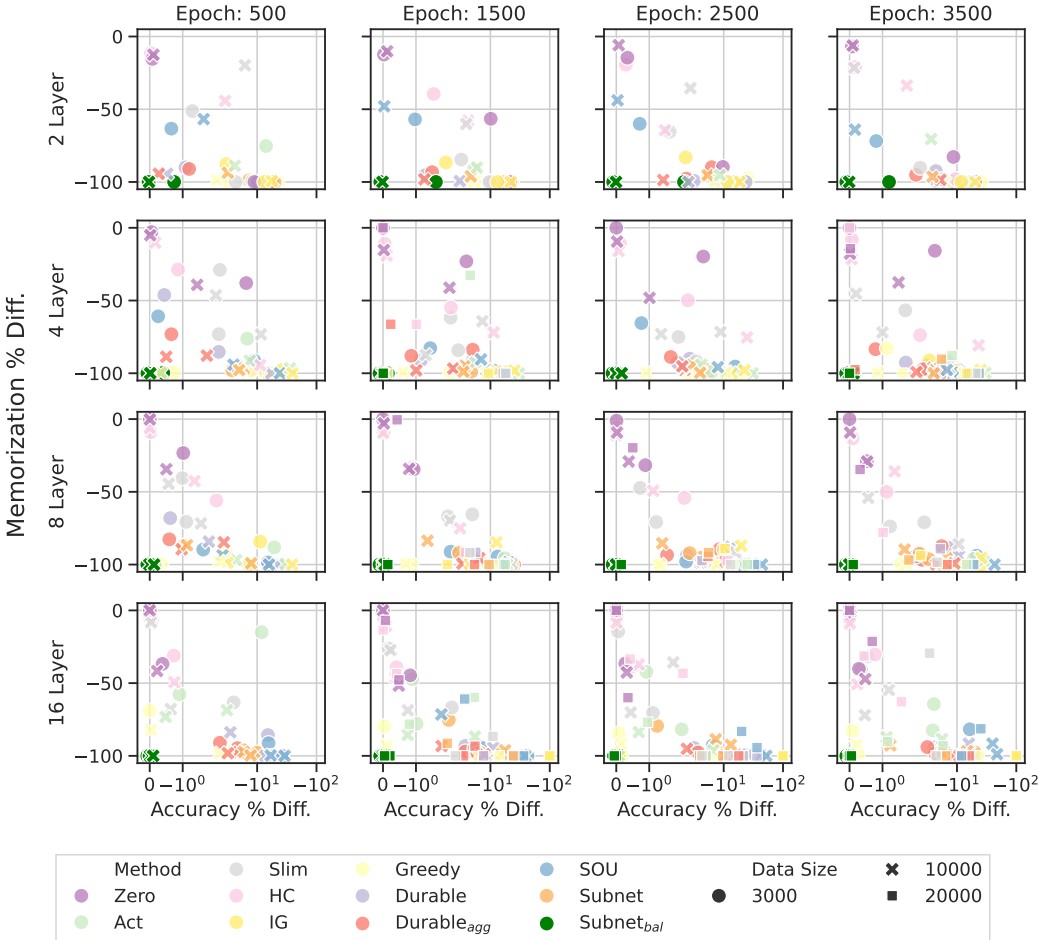

Figure 12: **Math + Noise**: Comparison of memorization % difference (closer to –100 better) and accuracy % different (closer to 0 better) before and after unlearning. Stratified by layers and training duration. Each point represents an average over three seeds. X-axis is on log scale. Aggregate results from both multiplication and increment models.

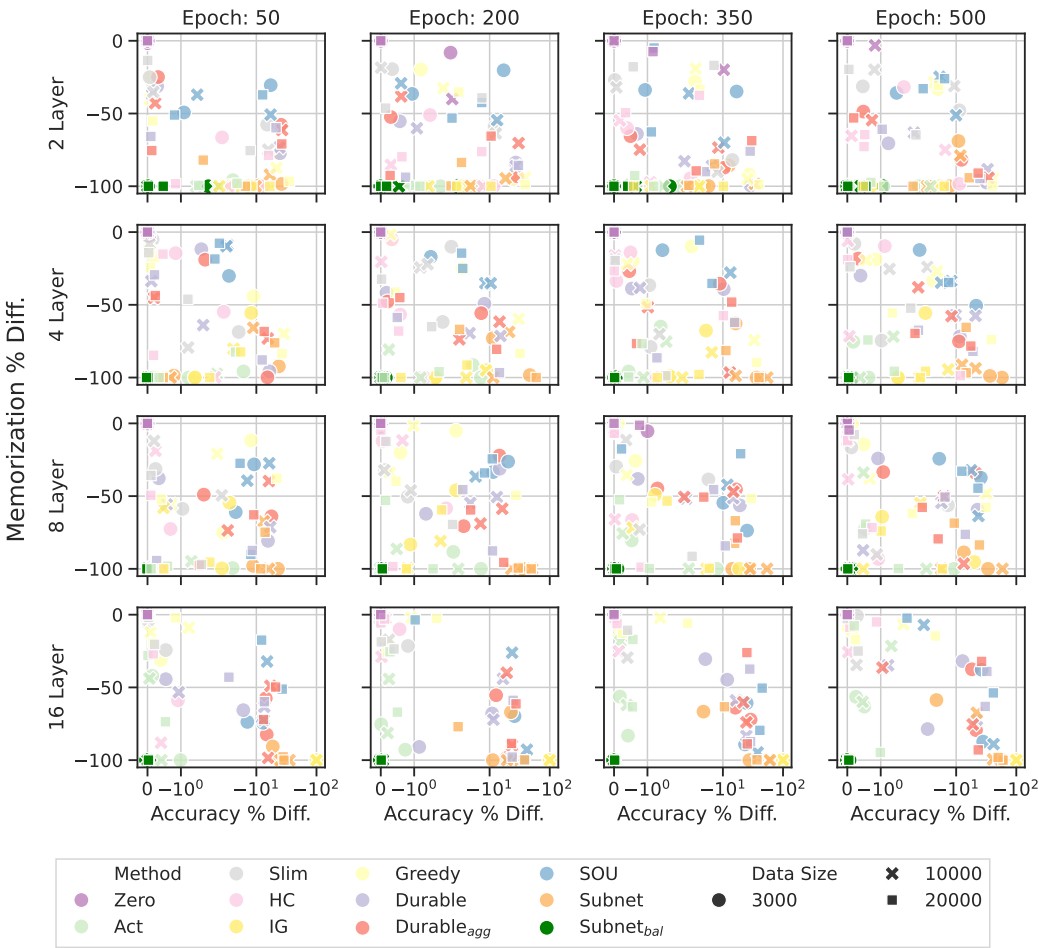

Figure 13: **Math + Backdoor** Comparison of memorization % difference (closer to –100 better) and accuracy % different (closer to 0 better) before and after unlearning. Stratified by layers and training duration. Each point represents an average over three seeds. X-axis is on log scale. Aggregate results from both multiplication and increment models.

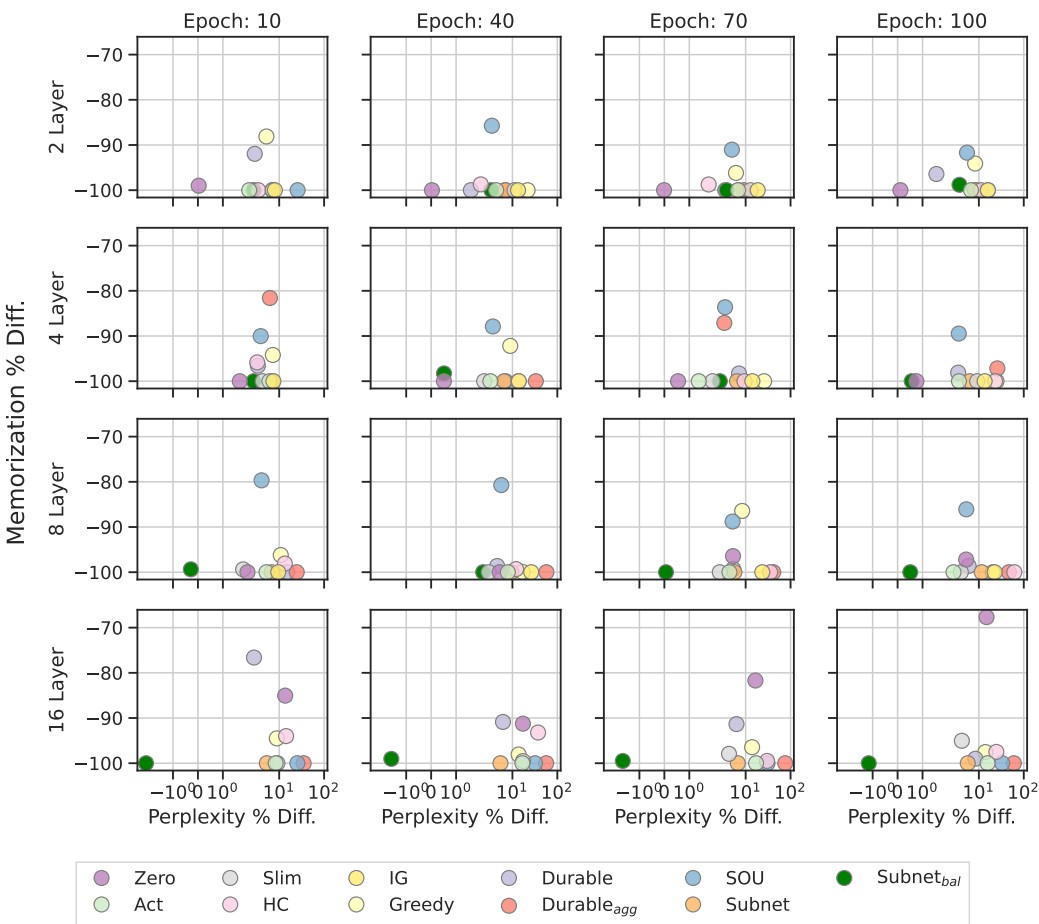

Figure 14: **Language + Noise:** Comparison of memorization % difference (closer to –100 better) and % difference in perplexity (closer to 0 better) before and after unlearning. Stratified by layers and training duration. Each point represents an average over three seeds. X-axis is on log scale.

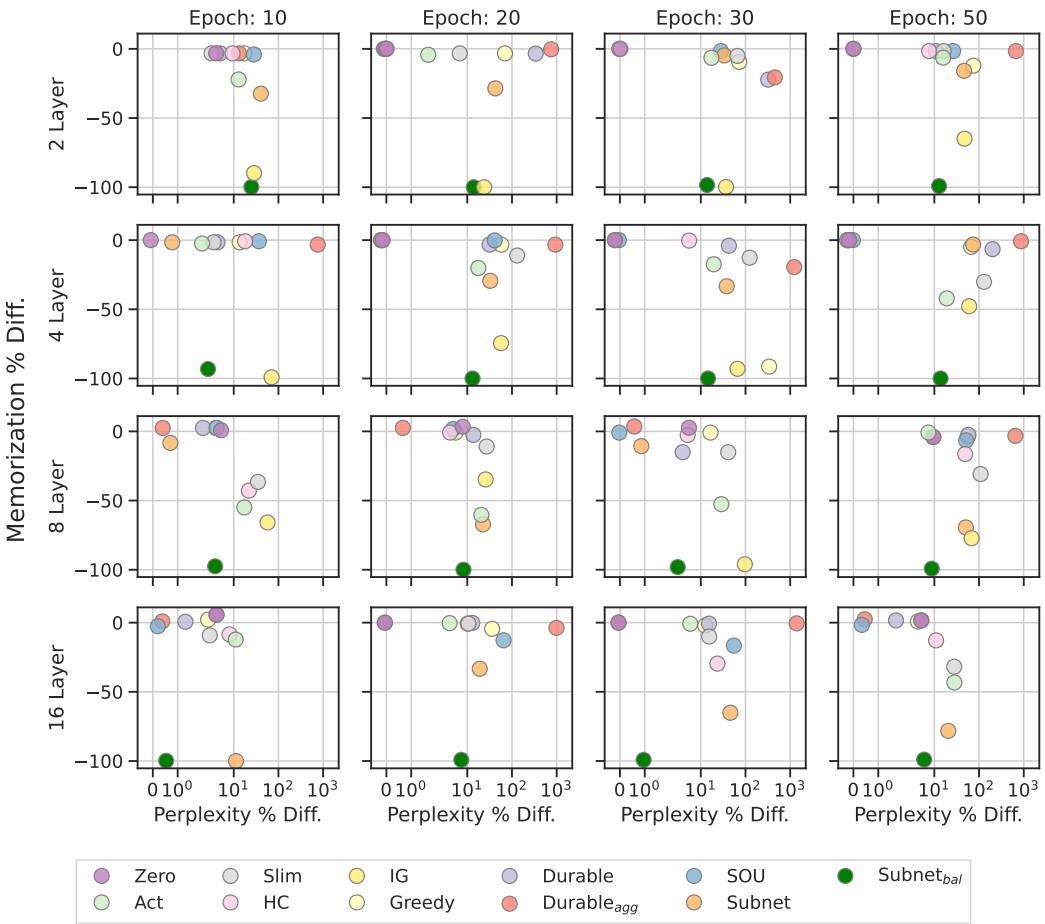

Figure 15: **Language + Backdoor:** Comparison of memorization % difference (closer to −100 better) versus % difference in perplexity (closer to 0 better) before and after unlearning. Stratified by layers and training duration. Each point represents an average over three seeds. X-axis is on log scale.

A.3.7 Why does **BalancedSubnet** work well?

Through extensive analysis in Tables 1, 2 & 3 and Figures 2, 3 & 5, we notice that **BalancedSubnet** consistently outperforms all other unlearning-based mitigation methods. In this section, we analyses the reasons for its success.

First we describe the design choices (Section A.3.4) that led to **BalancedSubnet**'s success:

1. **Weight-based:** From Fig. 2, we see that while both neuron- and weight-based methods can both preserve LM perplexity, weight-based methods substantially outperform neuron-based methods at mitigating memorization. This is likely due to weight-based methods operating at a finer level of granularity than neuron-based.

2. **1st order:** From Tables 1 & 2, we see that this first order method is much faster than **SOU**, a second order method. From Fig. 3, we see that this first order method outperforms both zero order methods (**Act**, **Zero**); this is likely due to first order information capturing more information about the loss landscape with respect to memorized content compared to zero order information.

3. **Non-iterative:** From Table 3, we see that **BalancedSubnet** is substantially faster than iterative methods like **Greedy**.

4. **Model-wise Scope:** From Tables 1, 2 & 3 and Fig. 3, we see that **BalancedSubnet** outperforms methods with layer-wise scopes (**Durable**, **SOU**). This is likely because layer-wise methods only rank weight importance w.r.t. other weights in the same layer. This means that some weights may be artificially under or over ranked due to the lack of model-wide weight importance comparison.

5. **Optimization-based:** From Fig. 5, we experimentally observe that by learning weight importance scores w.r.t memorization via optimization-based approaches, like **Subnet** and **BalancedSubnet**, we are able to more precisely localize and ablate the set of weights responsible for memorization and not responsible for general sequence generation. **BalancedSubnet**, like **Subnet**, is an optimization-based technique rather than a threshold-based technique. Threshold-based techniques assign proxy importance scores to each weight w.r.t. memorization and ablate weights with the highest scores. For example, gradients accumulated over the memorized set can be a proxy score for weight importance; weights with high proxy scores are ablated. While threshold-based techniques are more human-interpretable, the proxy metrics they rely on may not capture the nuanced dependencies within the LM weights that are responsible for memorization.

6. **Dual Objective:** The dual optimization objective in **BalancedSubnet** is essential to both remove memorized content while retaining strong performance on non-memorized data. Without the dual optimization objective, **BalancedSubnet** is equivalent to the **Subnet** strategy (which only optimizes to reduce memorization). On evaluating **Subnet**, we observed that, in some cases, it aggressively removed weights that were instrumental for generating non-memorized sequences. **BalancedSubnet** was, therefore, designed to also include the objective of minimizing loss on a held out "retain" set. By adding this second "retain" set objective into our loss function, we validate this hypothesis and report the increase in performance on an unseen test set (see Tables 1, 2 & 3).

Next we analyze the ratio of neurons/weights each unlearning method ablates from an LM in Figures 16, 18, 17, 19. We notice that most neuron-based methods prune $\sim 4-6\%$ of model neurons; when stratified by model size & training data (Fig. 16 & Fig. 18), we do not notice any clear trends in the TinyMem models. We notice most weight-based methods prune $\sim< 5\%$ of model weights with the notable exceptions of **Subnet** and **BalancedSubnet** for TinyMem models. We notice most weight-based methods prune $\sim< 0.01\%$ of weights with the notable exceptions of **Subnet** and **BalancedSubnet** for Pythia 2.8/6.9B models which prune $\sim 2-10\%$. For the TinyMem models, when stratified by model size (Fig. 17), we notice that both **Subnet** and **BalancedSubnet** prune more weights for larger models. For the TinyMem models, when stratified by model training data+artifact type (Fig. 19), we notice that both **Subnet** and **BalancedSubnet** roughly prune the most to least weights in the order of "Math+Noise", "Math+Backdoor", "Language+Backdoor", "Language+Noise". We speculate that both **Subnet** and **BalancedSubnet** pruned more weights when the model is inherently more sparse, as may be the case with an over-parameterized model (bigger model size or easier training task).

**BalancedSubnet**, the most performant method, pruned significantly more weights than any other method. This finding gave further credence to the dual-optimization-based design of **BalancedSubnet**. Despite dropping significantly fewer weights than **BalancedSubnet**, all other unlearning methods struggle to reach the same performance (memorization mitigation & performance preservation). This suggests that most methods are not able to precisely localize+ablate weights are responsible for memorized sequence generation/inconsequential to general sequence generation.

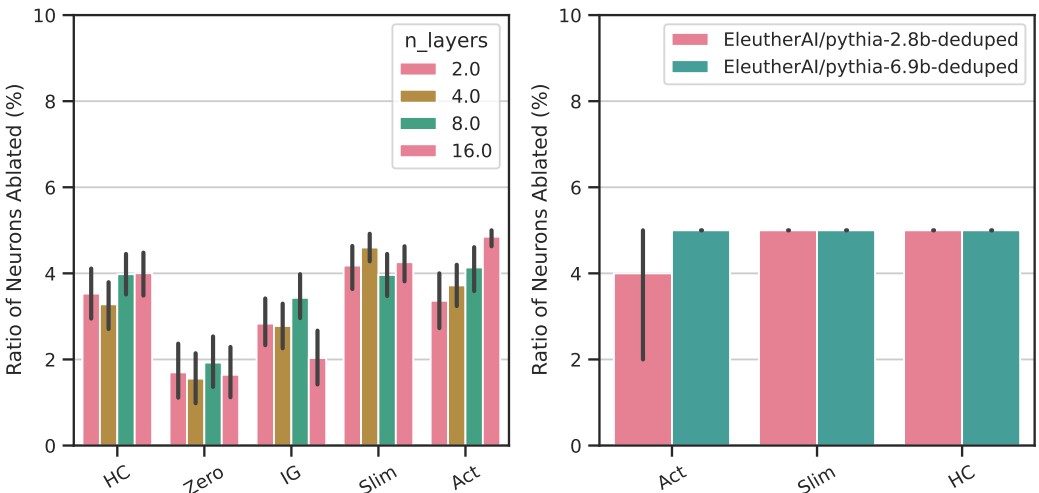

Figure 16: **Neuron Ratio Across Model Sizes.** Comparison of number of dropped neurons for neuron-based methods, for both `TinyMem` and Pythia for varying model sizes averaged across several unlearning times, data sizes, and three seeds.

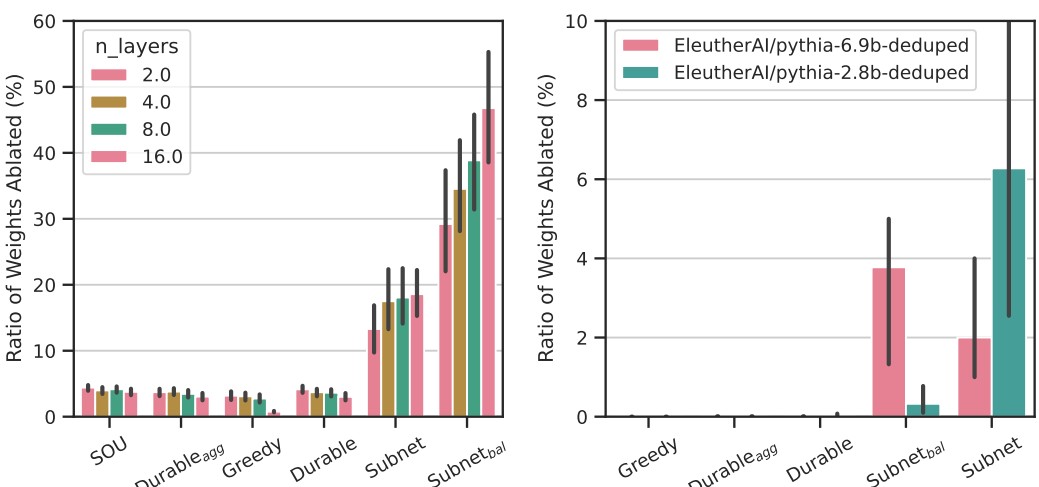

Figure 17: **Weight Ratio Across Model Sizes.** Comparison of number of dropped weights for weight-based methods, for both `TinyMem` and Pythia for varying model sizes averaged across several unlearning times, data sizes, and three seeds.

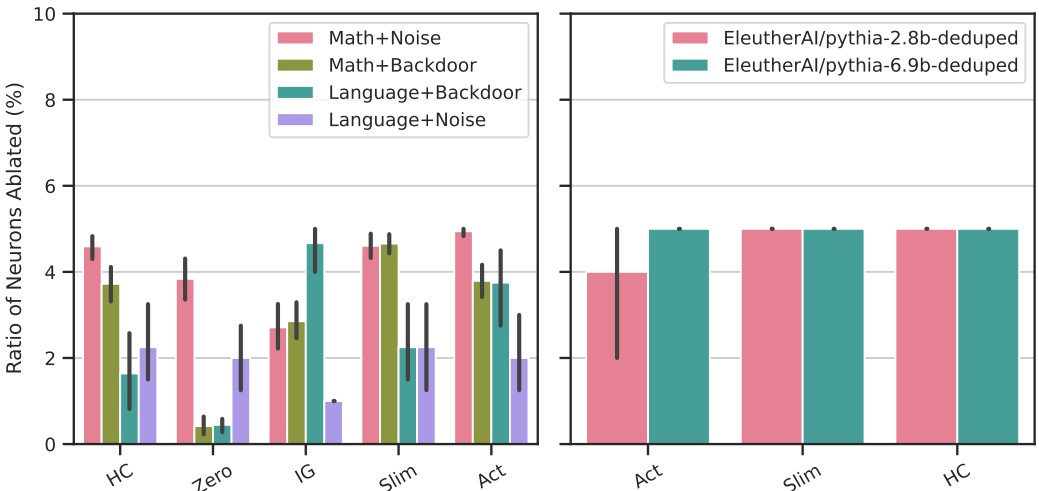

Figure 18: **Neuron Ratio Across Model Datasets+Artifacts.** Comparison of number of dropped neurons for neuron-based methods, for both `TinyMem` and Pythia for varying model types averaged across several unlearning times, data sizes, and three seeds.

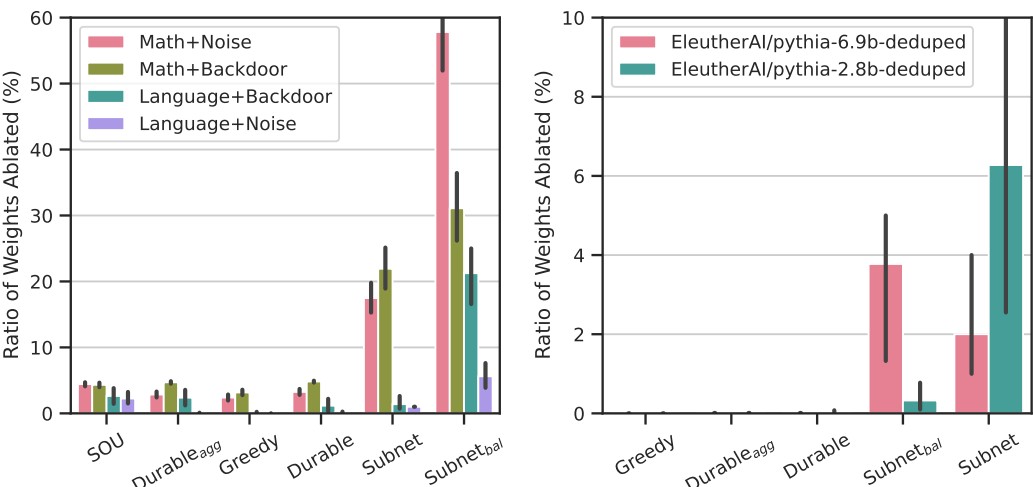

Figure 19: **Weight Ratio Across Model Datasets+Artifacts.** Comparison of number of dropped weights for weight-based methods, for both `TinyMem` and Pythia for varying model types averaged across several unlearning times, data sizes, and three seeds.

## A.4 PRODUCTION-GRADE UNLEARNING

### A.4.1 MEMORIZED DATA EXTRACTION FOR PYTHIA MODELS

We follow the methods outlined in Chang et al. (2024) to extract two datasets of memorized text, one each from Pythia 2.8B and 6.9B. Chang et al. (2024) considers a sequence to be memorized by an LM if it is able to produce completions that "nearly reconstruct" the original suffix, given an input prefix sequence; this definition is more relaxed than the one we consider in this work (Def 2.1), which deems a sequence to be memorized only if the LM can reconstruct the suffix verbatim using greedy decoding. Therefore, we analyze how many of the sequences from Chang et al. (2024) fit our definition of memorization, Def 2.1, when considering ($n = 72$, $k = 32$). Fig. 20 shows how memorization grows over training as model perplexity (a general metric of model performance) decreases (desirable). Memorization over training and perplexity over training for Pythia 2.8/6.9 can be visualized in Fig. 20. We evaluate Pythia model perplexity over 1632 randomly sampled sequences of the Pile (Gao et al., 2020), following the same random sampling procedure as (Chang et al., 2024).

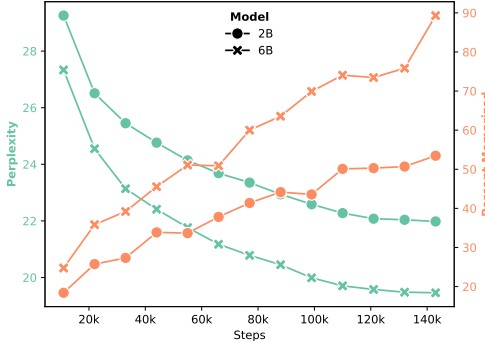

Figure 20: Perplexity and Memorization of Pythia models over training.

### A.4.2 PYTHIA: MACHINE UNLEARNING HYPER-PARAMETER SEARCH + SELECTION

**Hyper-parameter search:** For both Pythia 2.8/6.9B, we vary these hyperparameters for various methods:

**BalancedSubnet**:

- $ratio \in \{0.00001, 0.0001, 0.001, 0.01, 0.05, 0.1, 0.25, 0.3\}$
- $num\_epochs \in \{1, 10, 20\}$
- $loss\_weight \in \{0.9, 0.7, 0.5\}$

**Subnet**:

- $ratio \in \{0.00001, 0.0001, 0.001, 0.01, 0.05, 0.1, 0.25, 0.3\}$
- $num\_epochs \in \{1, 10, 20\}$

**HC**, **Slim**:

- $ratio \in \{0.00001, 0.0001, 0.001, 0.01, 0.05, 0.1\}$
- $num\_epochs \in \{1, 10, 20\}$

**Greedy**:

- $ratio \in \{0.00001\}$

**Act**:

- *ratio* $\in \{0.0001, 0.001, 0.01, 0.05, 0.1\}$

**Durable**, **Durable-agg**:

- *ratio* $\in \{0.00001, 0.0001, 0.001, 0.01, 0.05, 0.1\}$

**Selection Criteria:** For Pythia models, we scored the best run by assessing which training setting resulted in the lowest $score = M + P + t$ where $M$ = memorization percent difference before and after edit, $P$ = perplexity percent difference before and after edit, $t$ is an optional penalty to the score: if $M = 0$, then $t = 100$, else $t = 0$. We include this penalty to ensure that methods that do not reduce memorization at all are penalized more than methods that do reduce memorization to any extent. The lower the score, the better. We impose an inclusion criteria for each unlearning run: model perplexity has to be $< 500$.

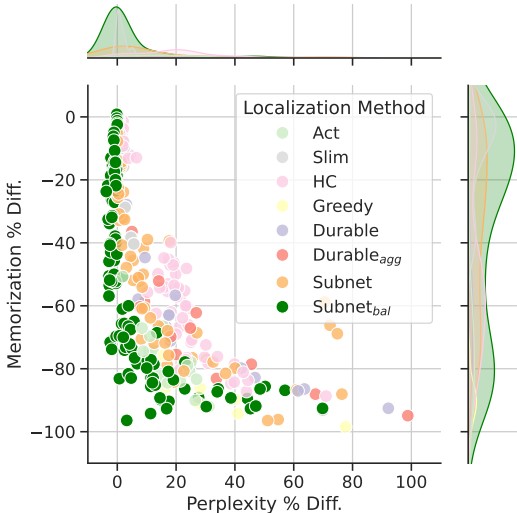

Figure 21: **Unlearning strategies comparison.** Comparison of memorization % difference (closer to –100 better) and % difference in perplexity (closer to 0 is better), before and after unlearning. We visualize all unlearning results with model perplexity $< 40$ and % memorized $< 50$. We notice that **BalancedSubnet** (Subnet$_{bal}$) has the highest density both near -100% difference in % memorized and 0% difference in perplexity out of all of the unlearning methods.

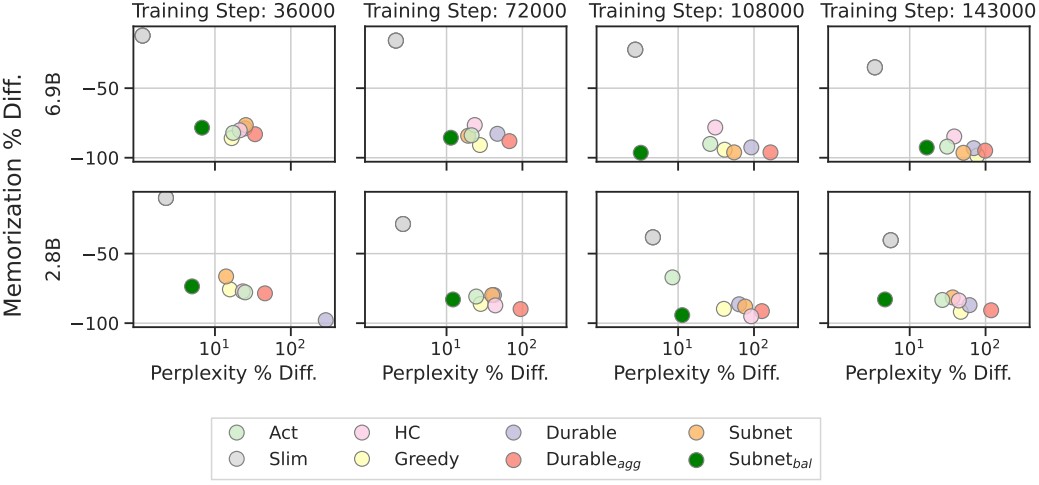

Figure 22: **Pythia Comparison** of memorization % difference (closer to −100 better) versus perplexity % different (closer to 0 better) before and after unlearning. Stratified by model type and train step. X-axis is on log scale.

Table 6: **System Parameters** for Polaris (ALCF, 2024) and Perlmutter (NERSC, 2024) to estimate energy and carbon usage.

| Machine | $c_{uf}$ | $c_{tdp}$ | $ngpu$ | $g_{uf}$ | $gpu_{tdp}$ | $DRAM$ | $PUE$ |
|---|---|---|---|---|---|---|---|
| Polaris | 0.5 | 225 | 4 | 1 | 250 | 512 | 1.58 |
| Perlmutter | 0.5 | 280 | 4 | 1 | 300 | 256 | 1.58 |

Table 7: **Energy & Carbon Estimates** for our experiments based on Eq. (4) & Eq. (5) respectively.

|  | Experiment | Machine | Node Hours | Energy (kWh) | Carbon (Kg) |
|---|---|---|---|---|---|
| (i) | Training TinyMem models | Polaris | 1116 | 2297 | 910 |
| (ii) | TinyMem Mitigation | Polaris | 6251 | 12,871 | 5097 |
| (iii) | Production-Grade Mitigation | Perlmutter | 726 | 1,647 | 430 |
| | All Experiments (total) | - | - | 16,815 | 6,437 |

## A.5 Loss Landscape Visualization

We created the loss landscape visualizations in Fig. 4 following the approach introduced by Li et al. (2018). We use the corresponding Python package which can be found at `https://github.com/marcellodebernardi/loss-landscapes`.

## A.6 Computation Cost of Experiments

The experiments in this paper used approximately **16815 kWh of energy, and 6437 Kg of carbon**.

We detail the computational resource (i.e., node hours, energy, carbon) used by our experiments below.

To calculate node hours, we time all of our final experiments and *triple that value* to account for extra debugging time, faulty runs, and any unaccounted for compute usage.

To calculate energy usage and carbon cost of our experiments, we follow the methodology detailed by Bouza et al. (2023):

$$energy = NH * ((c_{\text{uf}} * c_{\text{tdp}}) + (ngpu * g_{\text{uf}} * gpu_{\text{tdp}}) + (0.3725 \text{ W/Gb} * DRAM)) * PUE, \quad (4)$$

where *NH* is node hours, $c_{\text{uf}}$ is the CPU usage factor, $c_{\text{tdp}}$ is the CPU's thermal design power, *ngpu* is the number of GPUs on a node, $g_{\text{uf}}$ is the GPU usage factor (we assume 100% utilization), $g_{\text{tdp}}$ is the GPU's thermal design power, *DRAM* is dynamic random access memory, and *PUE* is the power usage efficiency. *energy* is reported in watt hours. We record system-specific parameter values in Table 6.

$$carbon = (energy/1000) * CI \quad (5)$$

Above in Eq. (5), *energy* is obtained in watt hours from Eq. (4), and *CI* is the carbon intensity reported based on the yearly regional average for each computing center from (ElectricityMaps, 2024). *carbon* is reported in grams.

To offset the carbon cost of these experiments, we publicly release our check pointed trained models (upon publication).

We group experiments into three main phases: *(i)* Training TinyMem Models; *(ii)* TinyMem Models LM Mitigation and *(iii)* Production-Grade LM Memorization Mitigation. We report the resource usage for each phase of experimentation in Table 7.

### A.6.1 Energy and Carbon Savings due to TinyMem

In Table 8 we estimate the per-experiment node hour, energy, and carbon usage for both phase (ii) which used TinyMem to comprehensively test our mitigation methods and phase (iii) which extended

Table 8: **Experiment-Wise Resource Usage.** We calculate the number of experiments for phase (ii) and phase (iii) from Section A.3.5 & Section A.4.2 respectively. We then calculate the per experiment Node Hours (NH), Energy, and Carbon usage using estimates from Table 7.

| Experiment | # Experiments | NH/Exp. | Energy /Exp. (kWh) | Carbon/Exp. (Kg) |
|---|---|---|---|---|
| (ii) TinyMem | 102698 | 0.06 | 0.13 | 0.05 |
| (iii) Production-Grade | 1216 | 0.60 | 1.35 | 0.35 |

Table 9: **Hypothetical Phase (ii) Resource Usage.** Resource usage estimates of doing $102,698$ phase (ii) experiments with TinyMem and without TinyMem (i.e, with production-grade LMs) using experiment-wise resource usage estimates from Table 8.

| Resource Type | w/ TinyMem | w/o TinyMem | % Increase |
|---|---|---|---|
| Node Hours | 6,251 | 61,619 | 986% ($\uparrow$) |
| Energy (kWh) | 12,871 | 138,642 | 1077% ($\uparrow$) |
| Carbon (Kg) | 5,097 | 36,316 | 712% ($\uparrow$) |

the most promising mitigation methods from TinyMem to production-grade models. Based on the results in Table 8, if we had to conduct the same level of experimentation in phase (ii) without the use of TinyMem (i.e., using production-grade models directly), we would use:

- 0.60 NH/Exp. * 102698 Exp. = **61619 Node Hours**

- 1.35 KWh/Exp. * 102698 Exp. = **138,642 kWh Energy**

- 0.35 Kg/Exp. * 102698 Exp. = **36,316 Kg Carbon**

We compare the actual experiment costs of phase (ii) with TinyMem with the hypothetical costs of phase (ii) without TinyMem in Table 9.

## A.7 EXTENDED RELATED WORK

**Knowledge Editing** aims to change specific facts, associations, or information embedded in an LM outside of the constraints of traditional model training. Model editing requires the ability to localize learned information within subsets of the weight space and employs efficient and targeted methods to change this information while mitigating its effects of other information also embedded in the weight space. Model editing can be used to remove or alter private information, incorrect information, outdated information, biased information, and harmful information stored within model weights (Wu et al., 2023; Yan et al., 2024; Chen et al., 2023; Wang et al., 2024). Model editing can enable machine learning models to more exactly reflect human knowledge, without the massive overhead cost of typical model pre-training/fine-tuning (Meng et al., 2023). Zhu et al. (2020) propose an approach to modify specific learned facts encoded withing a LM's weights, while preserving model performance on other previously learned knowledge via a constrained optimization problem. Dai et al. (2022) developed attribution methods to decipher which neurons are responsible for specific facts within LMs and developed methods to manipulate these neurons to edit a given fact. Cao et al. (2021) and Mitchell et al. (2022) both propose hypernetwork based approaches to edit facts within models. Hypernetworks are additional networks that are trained to predict which weights are responsible for a given fact and how to modify the weights of a given neural network to better represent the desired knowledge. Meng et al. (2022) proposed Rank-One Model Editing (ROME): by interpreting multi-layer perceptrons as key-values stores, ROME is able to replace specific keys-value pairs to override old or establish new knowledge associations in the model.

**Machine unlearning techniques are a subset of knowledge editing techniques.** Machine unlearning encompasses a broad class of techniques which aim to remove influence of a particular training data point from a trained machine learning model (Yao et al., 2024; Bourtoule et al., 2021; Cao & Yang, 2015). Machine unlearning is particularly important amidst recent legislation like GDPR (Voigt & Von dem Bussche, 2017) which mandate the "the right to be forgotten". The post-training localization and mitigation techniques we describe in this paper can be categorized as machine un-

learning techniques, as they aim to remove the influence of a class of datapoints from a model (namely memorized datapoints). In practice there are two broad sets of machine unlearning techniques: exact and approximate. Exact techniques guarantee that a data point is removed from a model's training objective (Bourtoule et al., 2021). Approximate techniques do not provide such guarantees but rather aim to empirically demonstrate that a model is not influenced by a data point (Yao et al., 2024; Chang et al., 2024; Stoehr et al., 2024; Maini et al., 2023).

