# OpenReview forum: "Mitigating Memorization in Language Models"
_ICLR.cc/2025/Conference — ICLR 2025 Spotlight_

### Official Review · Reviewer_osoQ · 2024-10-17

**Soundness:** 4
**Presentation:** 3
**Contribution:** 3
**Rating:** 8
**Confidence:** 2

**Summary:**

This paper focuses on addressing the challenge of language models (LMs) memorizing sensitive or private data during training, leading to potential issues such as data leakage. The primary contributions of the paper are:

1. TinyMem: The introduction of TinyMem, a suite of small, computationally efficient GPT2-style models that enable rapid prototyping and evaluation of memorization mitigation strategies.

2. Comprehensive Evaluation: An empirical evaluation of three main strategies for mitigating memorization: regularizer-based methods, fine-tuning-based methods, and machine unlearning-based methods, with five new unlearning-based approaches being proposed.

3. BalancedSubnet: Among the newly introduced methods, the BalancedSubnet unlearning method is highlighted for its superior performance in reducing memorization while preserving model performance across different datasets.

4. Applicability to Larger Models: The paper demonstrates that mitigation techniques developed using TinyMem can be successfully applied to production-grade models, such as the Pythia series, validating their effectiveness in real-world settings.

The paper provides a detailed analysis of these strategies, focusing on the trade-offs between mitigating memorization and maintaining model accuracy

**Strengths:**

1. This paper first introduces TinyMem, a new suite of lightweight models specifically designed for rapid testing of memorization mitigation strategies. Additionally, the proposal of five new unlearning-based methods, including the innovative BalancedSubnet approach, adds contribution to the field. This work also highlights a novel way of applying unlearning-based techniques to real-world, production-grade models, bridging the gap between research and application.

2. Extensive experiments involve 17 memorization mitigation strategies, including both existing and newly proposed methods. The experiments are carefully designed to compare regularizers, fine-tuning methods, and unlearning-based strategies under diverse training conditions. The comprehensive analysis across different model sizes, data types, and training durations demonstrates the robustness of the findings. The use of Pythia models to validate the transferability of the methods adds credibility to the results.

3. The paper is well-structured and clear, with a logical flow from the problem of memorization in LMs to the proposed solutions and experimental results. The introduction of key concepts such as TinyMem and BalancedSubnet is explained effectively, and the figures, while not in vector graphic format, are useful in illustrating results.

**Weaknesses:**

This paper seems to be missing some important unlearning baseline methods, e.g., Gradient Ascent + Descent and Gradient Ascent + KL divergence [1].

[1] Jin Yao, Eli Chien, Minxin Du, Xinyao Niu, Tianhao Wang, Zezhou Cheng, and Xiang Yue. 2024. Machine Unlearning of Pre-trained Large Language Models. In Proceedings of the 62nd Annual Meeting of the Association for Computational Linguistics (Volume 1: Long Papers), pages 8403–8419, Bangkok, Thailand. Association for Computational Linguistics.

**Questions:**

Can you describe the difference between the setting of mitigating memorization and knowledge editing? It seems like the gradient-based methods of these settings are similar.

---

> ### Author Response · Authors · 2024-11-16
>
> Thank you for your review. We are encouraged by the fact that you find our methodology to be rigorous!
>
> >  This paper seems to be missing some important unlearning baseline methods, e.g., Gradient Ascent + Descent and Gradient Ascent + KL divergence [1]
>
> Thank you for pointing out this interesting work! We have added it to our “Related Work” section. Since [1] was published in a peer-reviewed venue in August 2024 (less than 4 months prior to the submission deadline for ICLR 2025), we did not have time to include experimental comparisons of their proposed methods.
>
> We note that there are a few important study design differences between the our work and [1]: **Predicated on prior works that indicate that memorization is localizable [2, 3], we choose to only include unlearning strategies that focus on concept localization followed by concept erasure, rather than fine-tuning-based unlearning methods [1].** Therefore, the design of every unlearning strategy in our work is based on finding the top N neurons/weights that a model uses when producing a memorized string, followed by ablating these neurons/weights. In contrast, the unlearning strategies in [1] update every parameter of the neural network through a model fine-tuning scheme with the objective of gradient ascent on a “forget set” and gradient descent on a “retain set”. While [1]’s approach is very popular it has two major shortcomings:
> 1. Full gradient calculations for the entire network (full forward + backward pass) are computationally expensive, making [1]’s unlearning strategies computationally comparable to fine-tuning the model (which we see in tables 1&2 works really well at mitigating memorization, but is much slower than many of the unlearning methods we consider).
> 2. Full model parameter updates, as detailed in [1], do not directly enable a machine learning practitioner to gain insights into localizing memorized sequences inside model weights. We believe localization-first strategies in the context of removing unwanted phenomena may enable ML practitioners to better understand how/where a model failed, while may be useful in preventing/mitigating future unwanted behavior.
>
> [2] Maini, Pratyush, et al. ‘Can Neural Network Memorization Be Localized?’ Proceedings of the 40th International Conference on Machine Learning, edited by Andreas Krause et al., vol. 202, PMLR, 23--29 Jul 2023, pp. 23536–23557, https://proceedings.mlr.press/v202/maini23a.html. Proceedings of Machine Learning Research.
>
> [3] Chang, Ting-Yun, et al. ‘Do Localization Methods Actually Localize Memorized Data in LLMs? A Tale of Two Benchmarks’. Proceedings of the 2024 Conference of the North American Chapter of the Association for Computational Linguistics: Human Language Technologies (Volume 1: Long Papers), edited by Kevin Duh et al., Association for Computational Linguistics, 2024, pp. 3190–3211, https://doi.org/10.18653/v1/2024.naacl-long.176.
>
> > Can you describe the difference between the setting of mitigating memorization and knowledge editing? It seems like the gradient-based methods of these settings are similar.
>
> Thank you for asking this question, we are happy to discuss. We agree, the gradient-based methods of both settings are similar.
>
> At a high level, knowledge editing encompasses a broad set of techniques and aims to change specific facts, associations, or information embedded in a neural network outside of the constraints of traditional model training. Therefore, we consider machine unlearning-based memorized mitigation techniques to be a subset of knowledge editing techniques with a more constrained/specific goal of removing the influence of specific training sequences from the trained model.
>
> In response to your question, we have added a more detailed survey of knowledge editing and its relation to machine unlearning in an expanded related works section in Appendix A.7 due to space limitations in the main text. We have uploaded a revised PDF. Please let us know if this addresses the question.

---

### Official Review · Reviewer_phyr · 2024-10-30

**Soundness:** 2
**Presentation:** 2
**Contribution:** 2
**Rating:** 6
**Confidence:** 4

**Summary:**

This paper points out that developing and evaluating memorization mitigation strategies for open-sourced LLMs is a critical challenge due to the lack of memorized data or a big-time cost. To address these issues, the authors first propose a suite of GPT2-style models, namely TinyMem, to rapidly develop and evaluate memorization mitigation strategies. Based on TinyMem, this paper investigates three classes of existing methods, including training-time regularizer-based strategies, post-training fine-tuning-based strategies, and post-training unlearning-based strategies. Additionally, this paper also proposes five new unlearning-based methods. The experimental results show the effectiveness of these proposed methods.

**Strengths:**

1. This paper focuses on the critical problems, namely, the challenges of developing and evaluating memory elimination methods directly on large models.
2. This paper innovatively proposes a series of small GPT2-style models (TinyMem), which is used to conduct memorization injection and memorization mitigation experiments in a quick and computationally efficient way.
3. This paper proposes five new unlearning-based methods, and among them, BalancedSubnet outperforms other methods at removing memorized information while preserving performance on target tasks.

**Weaknesses:**

1. I am afraid that the evaluation results of different methods on TinyMem cannot truly reflect the performance of these methods in the memorization elimination of LLMs. On the one hand, the results in Table 1 and Table 2 show that some unlearning-based methods do not perform consistently in different types of tasks and memorization elimination. So how to effectively compare the comprehensive performance of different methods? On the other hand, the types of memorization within LLMs that need to be mitigated in practical applications are more complex and diverse. The two types of memorization (Noise and Backdoor) may be too simple compared to the real harmful memorization. The evaluation based on these two memorizations may hardly represent the ability of different methods to eliminate different types of memory within LLMs.
2. The innovation of the five proposed methods and their differences from previous methods should be introduced in detail. Additionally, although the algorithms are listed in the appendix, they still lack detailed introductions, which are difficult to understand.
3. Lack of detailed analysis of the reasons for the good performance of BalancedSubnet.

**Questions:**

1. In Line 066, "the few existing models with known memorized data are large" lacks detailed examples and references.
2. In lines 143-144, why test whether the next n−k tokens are matched with the clean sequence? Shouldn't it still match with the noise sequence? Or only disturb the first k words of one sequence?
3. Why is there a big difference in the mitigation performance and time of the same method in Nosie and Backdoor? Is there a difference in the difficulty of eliminating different types of memorization? The time consumption of the BalancedSubnet method in Table 1 is 0.00. How is it calculated here? Is there an error?
4. It is recommended to bold the best results in all tables.
5. The tenses in Lines 327-336 are not consistent.

---

> ### Author Response · Authors · 2024-11-16
>
> Thank you for your careful review!
>
> > I am afraid that the evaluation results of different methods on TinyMem cannot truly reflect the performance of these methods in the memorization elimination of LLMs.
>
> As we show in section 7, our results from TinyMem directly extend to production-grade LMs (Pythia 2.8B/6.9B) which are trained on the Pile (~800GB corpus).  In particular, the memorized data set we use in our mitigation experiments for both Pythia 2.8/6.9 B was indeed much more diverse and is from a public data set of memorized sequences [1]. Our results in Section 7 demonstrate a continuation of the trends we observed in our Tiny Mem experiments (Section 6): namely that BalancedSubnet offers superior unlearning performance at a moderate loss in perplexity and a competitive time cost in both TinyMem models and production-grade Pythia models.
>
> > The two types of memorization (Noise and Backdoor) may be too simple compared to the real harmful memorization. The evaluation based on these two memorizations may hardly represent the ability of different methods to eliminate different types of memory within LLMs.
>
> Further, as you point out, our TinyMem model suite has two potential limitations which we address in our study.
>
> 1. The size of our toy models are much smaller than models typically deployed in production and, therefore, may not be representative of larger models that display emergent behavior at scale. To mitigate this concern, we do extensive testing to ensure that memorization in TinyMem models scale with three key properties: model size, training data size, and data duplication. In Figure 6, we observed the same scaling laws in our TinyMem suite that production-grade models have displayed which is well documented in prior literature (See Appendix A.1.3 for details).
>
> 2. The nature of our tasks may not be representative of typical language model tasks, limiting our ability to generalize our results to real-world models. To combat this, we include both noise-based and backdoor-based memorization artifacts in TinyMem because noise-based artifacts are easy-to-learn and backdoor-based artifacts are hard-to-learn (see Figure 6). Further, we randomized the noising-strategy by sequence, meaning each noised sequence had a different noising pattern; thus making it more difficult for the model to learn different sets of noised sequences. Finally, we include two sets of models, those training on math sequences and those trained on a real language modeling task (Wikipedia). Finally, we vary model size and training time to assess if these variables affect mitigation method success. By including models with diverse artifacts with unique training properties and models training on different datasets, we test each mitigation method on a wide range of memorized sequences.
>
> Finally, we would like to emphasize that in addition to TinyMem: **Results from section 7 demonstrate that unlearning methods developed on TinyMem models successfully scale to production-grade memorization mitigation scenarios.**
>
> [1] Chang, Ting-Yun, et al. ‘Do Localization Methods Actually Localize Memorized Data in LLMs? A Tale of Two Benchmarks’. Proceedings of the 2024 Conference of the North American Chapter of the Association for Computational Linguistics: Human Language Technologies (Volume 1: Long Papers), edited by Kevin Duh et al., Association for Computational Linguistics, 2024, pp. 3190–3211, https://doi.org/10.18653/v1/2024.naacl-long.176.

---

> ### Author Response · Authors · 2024-11-16
>
> > Lack of detailed analysis of the reasons for the good performance of BalancedSubnet.
>
> Thank you for the review. We are encouraged that you found our experimental methodology to demonstrate the clear advantages of our proposed BalancedSubnet strategy!
>
> To address your concerns, we added extensive analysis of the insights that motivated each method in Appendix A.3.1. We experimentally evaluated the intuition that underlies these methods in Section 6 & 7. Further, we added a new Table 5 which compares each method along 9 dimensions. By juxtaposing the key differences between methods (Table 5) with our experimental results (Tables 1, 2, 3), we evaluated which methodological design features were most beneficial, detailed both at a high level in Appendix A.3.4, and on a detailed level w.r.t BalancedSubnet in A.3.7.
>
> We summarize three key practical insights that underlie the benefits of BalancedSubnet:
> 1. The dual optimization objective in BalancedSubnet is essential to both remove memorized content while retaining strong performance on non-memorized data. Without the dual optimization objective, “BalancedSubnet” is equivalent to the “Subnet” strategy (which only optimizes to reduce memorization). On evaluating “Subnet”, we observed that, in some cases, it aggressively removed weights that were instrumental for generating non-memorized sequences. BalancedSubnet was therefore designed to also include the objective of minimizing loss on a held out “retain” set. By adding this second “retain set” objective into our loss function, we validate this hypothesis and report the increase in performance on an unseen test set (see Tables 1, 2, 3).
> 2. Balanced subnet, unlike greedy, is not iterative meaning it is fast (see Tables 1, 2, 3).
> 3. BalancedSubnet, like Subnet, is an optimization-based technique rather than a threshold-based technique. Threshold-based techniques assign proxy importance scores to each weight w.r.t. memorization and ablate weights with the highest scores. For example, gradients accumulated over the memorized set can be a proxy score for weight importance; weights with high proxy scores are ablated. We experimentally find that by instead learning the importance scores of weight via optimization-based approaches, like Subnet and BalancedSubnet, we are much more likely to precisely localize and ablate the set of weights responsible for memorization and not responsible for general sequence generation (Figure 5).
>
> Finally, prompted by your review, we conducted further analyses and made an interesting observation in Figures 16,17,18,19: BalancedSubnet typically pruned ~20-60% LM weights, Subnet pruned ~5-20% of LM weights, and all other unlearning-based methods pruned ~<5% of LM weights. Interestingly, BalancedSubnet, the most performant method, pruned significantly more weights than any other method. This finding gave further credence to the dual-optimization-based, non-iterative design of BalancedSubnet. Despite dropping significantly fewer weights than BalancedSubnet, all other unlearning methods struggle to reach the same performance (memorization mitigation & performance preservation). This suggests that most methods are not able to precisely localize+ablate weights are responsible for memorized sequence generation/inconsequential to general sequence generation.

---

> ### Author Response · Authors · 2024-11-16
>
> > Why is there a big difference in the mitigation performance and time of the same method in Noise and Backdoor? Is there a difference in the difficulty of eliminating different types of memorization?
>
> Great question. **Yes, there is a difference in difficulty of eliminating different types of memorization.** We suspected this was the case when designing this study, and therefore included different types of memorized sequences. Based on the results in Figures 6.a and 6.c, we notice that it takes longer for models to memorize noised sequences compared to backdoored sequences in figures 6.b and 6.d. We therefore hypothesized that it was easier for models to learn backdoors vs. noise, and consequently we wondered if it is harder to remove easy-to-learn data vs. hard-to-learn data. As you correctly pointed out, in Tables 1 & 2, it turns out that there is indeed a substantial difference in performance with backdoors being harder to remove than noised sequences. This result reinforced our hypothesis that easy-to-learn sequences are harder to remove and hard-to-learn sequences are easier to remove. We describe this observation in lines 131-136. Because our intuition was supported by our experimental data in our toy models, we further found that the strategies that performed well on both noise and backdoors in the toy models were the same methods that performed best on production-grade Pythia models (e.g., BalancedSubnet). Please feel free to let us know if you have follow up questions.

---

> ### Author Response · Authors · 2024-11-16
>
> > The innovation of the five proposed methods and their differences from previous methods should be introduced in detail. Additionally, although the algorithms are listed in the appendix, they still lack detailed introductions, which are difficult to understand.
>
> Thank you for pointing this out. To rectify this confusion, we have added detailed descriptions to the Appendix A.3. For each method we include detailed “intuitions” that inspired the approach (A.3.1) & “technical descriptions” (A.3.2), and the “psuedo-code” algorithmic descriptions (A.3.3). We summarize “key differences” between all the methods in Table 5 and Appendix A.3.4 along 9 different features. We use the analysis done in Table 5 to further investigate the key reasons for BalancedSubnet’s success in Appendix A.3.7.
>
> > In Line 066, "the few existing models with known memorized data are large" lacks detailed examples and references.
>
> Thanks for catching this. We have updated and reuploaded the PDF to provide citations both in the initial claim, and a detailed follow up analysis of all the publicly available model sizes and their corresponding publicly available memorized sequence datasets in Appendix A.1.4. In summary, we note that TinyMem models range from (814K - 9.6M) traininable parameters, while still demonstrating the same memorization scaling properties as production grade models (which range from 125M - 66B parameters). Therefore, we argue that a small model test suite like TinyMem enables rapid prototyping and testing of memorization mitigation methods on a representative set of efficient models, which can then be tested on production-grade models without incurring the initial development cost. Further, we find that of the 3 studies which we are aware of that release their (model, memorized sequence) pairs, we were only able to successfully recover 2 of them. Evaluating if models have memorized data is prohibitively expensive for many organizations as the training dataset is not always transparent, nor is it always computationally feasible to iteratively assess LM memorization over a corpus like the Pile (~800 GB).
>
> > In lines 143-144, why test whether the next n−k tokens are matched with the clean sequence? Shouldn't it still match with the noise sequence? Or only disturb the first k words of one sequence?
>
> Good catch, this was a typo. We have corrected and reuploaded a revised PDF. To clarify, we actually prompt the model with the first n tokens from the clean sequence and we check if the next n-k tokens matched the noised sequence. We choose to measure memorization via this approach as prompting the model with noise directly resulted in ~100% memorization of noised sequences; it was easy for a model to generate the rest of a nosied sequence when prompted with the beginning of a noise sequence. On the other hand, we choose a more realistic scenario where an unknowing user might prompt a model with a clean sequence but the model may elicit a memorized noised completion that was present in the training set.
>
> > The time consumption of the BalancedSubnet method in Table 1 is 0.00. How is it calculated here? Is there an error?
>
> Thank you for catching this typographical error. The correct timing is “6.00” seconds. We have meticulously checked the results from all of our tables and confirmed there are no other typographical errors. To address how we timed the experiments, we calculated wall-clock runtime using Python’s “time” module for each experiment averaged over 3 seeds. We have fixed this typo and uploaded a revised PDF.
>
> > The tenses in Lines 327-336 are not consistent.
>
> Thank you for catching this inconsistency, we have updated the language to use a consistent tense (and uploaded a revised PDF).

---

> ### Author Response · Authors · 2024-11-20
>
> We would like to check in to see if you found our additional paper contributions useful to address your comments. If so, can you update your score accordingly? If not can you please be more specific as to what may be helpful for us to address during the rebuttal period? We would be glad to clarify any additional details.

---

> > ### Comment · Reviewer_phyr · 2024-11-25
> >
> > Thanks for your response. Most of my concerns have been addressed and I will adjust my score.

---

> > > ### Author Response · Authors · 2024-11-26
> > >
> > > Thank you for your review! We are happy to clarify any remaining concerns.

---

### Official Review · Reviewer_zLBY · 2024-11-06

**Soundness:** 3
**Presentation:** 2
**Contribution:** 2
**Rating:** 8
**Confidence:** 4

**Summary:**

This manuscripts presents a comparison of 17 methods for reducing memorization in large language models (LLMs) as well as a testbench
 for measuring model memorization comprising of a set of small models called TinyMem. The authors find that the BalanceSubnet method is the most effective at reducing model memorization while preserving performance.

**Strengths:**

The authors present a comprehensive comparison of 17 methods for reducing memorization.

Applying the TinyMem test bench to larger models (Pythia) demonstrates the generalizability of the result

**Weaknesses:**

No major methodological weaknesses, however I find this study to be a straightforward comparison of several (existing and new) techniques without any particular novel insights. While this type of work is useful, the authors should extend the work by investigating why the best method works well or extracting practical insights from the results (beside stating that the method is most effective).

**Questions:**

What does the test accuracy in Table 1 represent?
The perplexity scores in Table 2&3 are quite high - can you describe how these are calculated and on what dataset?

---

> ### Author Response · Authors · 2024-11-16
>
> Thank you for the review. We are encouraged that you found our experimental methodology to be robust and that it demonstrates the clear advantages of our proposed BalancedSubnet strategy!
>
> >  the authors should extend the work by investigating why the best method works well or extracting practical insights from the results (beside stating that the method is most effective)
>
> To address your concerns, we added extensive analysis of the insights that motivated each method in Appendix A.3.1. We experimentally evaluated the intuition that underlies these methods in Section 6 & 7. Further, we added a new Table 5 which compares each method along 9 dimensions. By juxtaposing the key differences between methods (Table 5) with our experimental results (Tables 1, 2, 3), we evaluated which methodological design features were most beneficial, detailed both at a high level in Appendix A.3.4, and on a detailed level w.r.t BalancedSubnet in A.3.7.
>
> We summarize three key practical insights that underlie the benefits of BalancedSubnet:
> 1. The dual optimization objective in BalancedSubnet is essential to both remove memorized content while retaining strong performance on non-memorized data. Without the dual optimization objective, “BalancedSubnet” is equivalent to the “Subnet” strategy (which only optimizes to reduce memorization). On evaluating “Subnet”, we observed that, in some cases, it aggressively removed weights that were instrumental for generating non-memorized sequences. BalancedSubnet was therefore designed to also include the objective of minimizing loss on a held out “retain” set. By adding this second “retain set” objective into our loss function, we validate this hypothesis and report the increase in performance on an unseen test set (see Tables 1, 2, 3).
> 2. Balanced subnet, unlike greedy, is not iterative meaning it is fast (see Tables 1, 2, 3).
> 3. BalancedSubnet, like Subnet, is an optimization-based technique rather than a threshold-based technique. Threshold-based techniques assign proxy importance scores to each weight w.r.t. memorization and ablate weights with the highest scores. For example, gradients accumulated over the memorized set can be a proxy score for weight importance; weights with high proxy scores are ablated. We experimentally find that by instead learning the importance scores of weight via optimization-based approaches, like Subnet and BalancedSubnet, we are much more likely to precisely localize and ablate the set of weights responsible for memorization and not responsible for general sequence generation (Figure 5).
>
> Finally, prompted by your review, we conducted further analyses and made an interesting observation in Figures 16,17,18,19: BalancedSubnet typically pruned ~20-60% LM weights, Subnet pruned ~5-20% of LM weights, and all other unlearning-based methods pruned ~<5% of LM weights. Interestingly, BalancedSubnet, the most performant method, pruned significantly more weights than any other method. This finding gave further credence to the dual-optimization-based, non-iterative design of BalancedSubnet. Despite dropping significantly fewer weights than BalancedSubnet, all other unlearning methods struggle to reach the same performance (memorization mitigation & performance preservation). This suggests that most methods are not able to precisely localize+ablate weights are responsible for memorized sequence generation/inconsequential to general sequence generation.

---

> ### Author Response · Authors · 2024-11-16
>
> > What does the test accuracy in Table 1 represent? The perplexity scores in Table 2&3 are quite high - can you describe how these are calculated and on what dataset?
>
> Thank you for the review and question! We have updated Appendix A.1.1 and A.4.1 to contain further details about how/why we calculate accuracy and perplexity.
>
> > What does test accuracy represent
>
> Given a prompt P where P is of length N+K, test accuracy in table 1 represents the token-wise accuracy between G, the greedy decoding of K generated tokens from the model given a prompt P[:N], and the baseline sequence P[N:K]. For example, if the prompt=[2,4,6], and the baseline sequence continues=[8,10,12], and the model greedily generates=[8,11,12], then the accuracy would be 66%. For the math sequences, we choose to report the accuracy rather than the perplexity as there is a notion of “correctness” in mathematical sequences that is not present for general language modeling and we felt that accuracy is a more exact measure of a sequential math model compared to perplexity.
>
> > How we calculate perplexity
>
> We calculate the mean of the exponentiated negative log likelihood over the generated tokens. We follow this standard procedure:https://huggingface.co/docs/transformers/en/perplexity
>
> > Table 2
>
> Displays results of a model trained on a wikipedia dataset, therefore, we assess perplexity on a held-out wikipedia test dataset with 4.3 K sequences.
> - Train:https://huggingface.co/datasets/Salesforce/wikitext/viewer/wikitext-103-raw-v1/train
> - Test:https://huggingface.co/datasets/Salesforce/wikitext/viewer/wikitext-103-raw-v1/test
> - As these sets of models are trained on a relatively small corpus (1.8M sequences), it is expected that the perplexities would be higher than that of production grade models trained on the pile (e.g., Pythia).
>
> >Table 3
>
> Displays results of a pre-trained Pythia 2.8/6.9B. We follow the sampling procedure in [1] and calculate the perplexity of the model on randomly sampled 1632 sequences of the Pile. To see how perplexity of both models evolves over training on this dataset, please refer to Figure 20.
>
> **We would like to emphasize that the effectiveness of a memorization mitigation strategy is relative to the performance of the baseline model, which is why we additionally report percent increases/decreases in perplexity in Figures 3 and 5.**
>
> [1] Chang, Ting-Yun, et al. ‘Do Localization Methods Actually Localize Memorized Data in LLMs? A Tale of Two Benchmarks’. Proceedings of the 2024 Conference of the North American Chapter of the Association for Computational Linguistics: Human Language Technologies (Volume 1: Long Papers), edited by Kevin Duh et al., Association for Computational Linguistics, 2024, pp. 3190–3211, https://doi.org/10.18653/v1/2024.naacl-long.176.

---

> ### Author Response · Authors · 2024-11-16
>
> > however I find this study to be a straightforward comparison of several (existing and new) techniques without any particular novel insights
>
> **We would like to emphasize a core contribution: the TinyMem model suite which enabled more rapid exploration of memorization mitigation methods than previously possible with production-grade LMs (detailed in Appendix A.1.4).** To summarize:
> 1. TinyMem models are substantially smaller (814K - 9.6M traininable parameters) than their production-grade models (which range from 125M - 66B parameters) which enabled more rapid inference and gradient-based procedures necessary for methods development workflows.
> 2. We performed extensive testing to ensure that memorization in TinyMem models scaled with three key properties known to be observed in production-grade models: model size, training data size, and data duplication (Figure 6/Appendix A.1.3).
> 3. We will release known memorized sequences in TinyMem which alleviates a substantial computational barrier of iteratively searching for memorized sequences in production-grade models with “internet-scale” datasets (e.g., Pile (~800 GB)).
>
> **Having a sound experimental methodology, as you note in your review, would be infeasible without TinyMem.** Using TinyMem: we rapidly tested regularizers, fine-tuning, & unlearning mitigation methods, did extensive method hyper-parameter tuning, developed 5 new mitigation strategies (each of which addressed shortcomings of the prior), and stress tested each mitigation method w.r.t. different model sizes, training data, training lengths, and memorized artifact types.  **We directly extended the most promising methods from TinyMem to production-grade LMs, which demonstrated TinyMem’s utility in modeling large LMs with the advantage of huge computational resource savings.** If we had done the same extensive testing in our study with production grade models instead of TinyMem, we would have used 986% more node hours, 1077% more energy, 712% more carbon (see Appendix A.6.1).

---

> ### Author Response · Authors · 2024-11-20
>
> We would like to check in to see if you found our additional paper contributions useful to address your comments. If so, can you update your score accordingly? If not can you please be more specific as to what may be helpful for us to address during the rebuttal period? We would be glad to clarify any additional details.

---

> > ### Comment · Reviewer_zLBY · 2024-11-23
> > **thanks for the detailed response.**
> >
> > The additional analyses submitted by the authors pushes this work above the acceptance threshold in my opinion. I have changed my score to reflect this.

---

### Meta-Review · Area_Chair_65CC · 2024-12-13

**Metareview:**

This work investigates methods to mitigate memorization in language models. It introduces *TinyMem*, a framework designed to rapidly develop and evaluate memorization mitigation strategies. The study finds that regularizers are slow and ineffective, fine-tuning is costly but effective, and unlearning methods are fast and precise at removing memorized data while preserving overall model performance. Consequently, the authors propose five other unlearning methods, among which *BalancedSubnet* performs the best.

The comparisons are comprehensive, and the results demonstrate the generalizability of the proposed methods.  It might be beneficial for the authors to incorporate key revisions from the Appendix into the main body during the rebuttal, as the current version may distract readers.

My final recommendation is acceptance.

**Additional Comments On Reviewer Discussion:**

Most of the confusion stems from the experimental setup, some misinterpretations of the paper, and aspects of its writing. The authors effectively address the reviewers' concerns.

---

### Decision · Program_Chairs · 2025-01-22

Accept (Spotlight)